# The complete trans-series for conserved charges in the Lieb–Liniger model

Zoltán Bajnok[1], János Balog[1], Ramon Miravitllas[1], Dennis le Plat[1], and István Vona[1,2]

[1]HUN-REN Wigner Research Centre for Physics , Konkoly-Thege Miklós u. 29-33, 1121 Budapest, Hungary
[2]Roland Eötvös University, Pázmány Péter sétány 1/A, 1117 Budapest, Hungary

[*bajnok.zoltan, balog.janos, ramon.miravitllas.mas, dennis.leplat, vona.istvan*]*@wigner.hun-ren.hu*

April 30, 2025

## Abstract

We determine the complete trans-series solution for the (non-relativistic) moments of the rapidity density in the Lieb–Liniger model. The trans-series is written explicitly in terms of a perturbative basis, which can be obtained from the already known perturbative expansion of the density by solving several ordinary differential equations. Unknown integration constants are fixed from Volin's method. We have checked that our solution satisfies the analytical consistency requirements including the newly derived resurgence relations and agrees with the high precision numerical solution. Our results also provides the full analytic trans-series for the capacitance of the coaxial circular plate capacitor.

# 1   Introduction

Two dimensional integrable models play special roles in many branches of physics. In particle physics they serve as toy models, where strongly interacting, non-perturbative effects can be tested in simplified circumstances. In some fortunate situations they can even solve four dimensional quantum gauge theories exactly [1]. In statistical physics they provide paradigmic models where fundamental questions related to phase transitions, thermalization or non-equilibrium dynamics can be exactly analysed. Besides the theoretical interest they can also describe strongly anisotropic solid state systems and show up in cold atom experiments [2, 3].

One of the simplest interacting integrable models is the Lieb–Liniger model [4], which describes the pointlike interaction of one dimensional bosons. This model served as a guiding example of statistical physics, being simple enough to be exactly soluble and complicated enough to describe non-trivial, in some cases even typical behaviour. This model was the first, where new ideas and methods were developed and tested the first time. The model was solved by Bethe ansatz and the ground-state energy in the thermodynamic limit was determined in terms of a linear integral equation [4] supporting Bogulyubov's theory. The finite temperature properties in terms of the thermodynamic Bethe ansatz were developed in [5], which sparked a lot of interest and research. The investigation of other observables, such as correlation functions, initiated fundamental developments and an arsenal of new methods [6], which were later used in many other circumstances. The Lieb–Liniger model is not only a useful toy model of academic interests, but it can be actually realized in cold atom experiments [2, 3].

One of the main observables is the rapidity density in the zero temperature groundstate, which satisfies a linear integral equation. Its zeroth moment provides the density, second moment gives the groundstate energy density, while higher moments are related to the expectation value of higher conserved charges. The rapidity density also controls the asymptotic behaviour of the correlation functions [7, 8]. Surprisingly, the very same quantity is directly related to the surface charge density of the coaxial circular plate capacitor [9]. It also appears in the relativistic $O(3)$ sigma model, in the case when the model is coupled to an external field. It describes the rapidity density in the groundstate, when the field is large enough and particles condense [10].

There is no hope to solve analytically the linear integral equation for the rapidity density. It is straightforward, however to perform a low density expansion. The resulting series has a finite radius of convergence and gives a good approximation for modest densities. The large density expansion of the rapidity density is a notoriusly difficult problem, which was achieved only recently [11, 12] based on the pioneering results of Volin for the $O(N)$ non-linear sigma models [13, 14]. This expansion, however is only asymptotic, which signals non-perturbative, exponentially suppressed corrections. The complete answer for the physical quantities is a double series, i.e. a transseries, which goes in the perturbative and the non-perturbative corrections. These corrections were analysed in details for *relativistic* observables in the related $O(3)$ model in [15, 16, 17, 18, 19, 20, 21]. The aim of the present paper is to develop the analysis of the *non-relativistic* observables to the same depth. These include the complete determination of the trans-series for the moments of the rapidity density and the investigation of its properties.

The paper is organized as follows: in Section 2 we revise the perturbative large density expansion in terms of the Fermi momentum of the particles, and generalize it to the higher

moments of the integral equation. For this we use the so-called running coupling variable [17] that eliminates unwanted logarithms in the series expansions. Next, we introduce another set of observables in Section 3 that were useful in solving this problem for the relativistic models [18, 21, 22] and relate them to the non-relativistic moments mentioned above via differential equations. Further, we discuss the solution of these ODEs and a way to fix the yet undetermined integration constants. In Section 4 we present the trans-series solution developed for the relativistic observables and argue that their structure and the differential relations of Section 3 together can be used to deduce a similar trans-series solution for the non-relativistic moments. Although our argument does not constitute as an analytic proof, we rigorously show in Subsection 4.2 (relegating the technical details to Appendix A) that the non-perturbative structure of our solutions is fully compatible with the system of ODEs in Section 3. Further, it leads to resurgence relations similar to those found in the relativistic case (see Subsection 4.3). At this point we leave the mathematical treatment and turn to the physical applications. First, as a by-product, in Section 5 we connect the trans-series solution of the lowest moment to a related historical problem in classical electrostatics, namely the capacity of the circular parallel plate capacitor. To our best knowledge, it is our work alongside [22] that provides the complete analytic expansion of this quantity for small separation of the disks. After this detour we re-express the energy density and higher moments of the Lieb–Liniger model in terms of an appropriately chosen and measurable expansion variable $g$ in Section 6. The latter is related to the number density of the particles and is commonly used in the literature. We validate our results numerically up to high precision in Section 7. We first confirm the resurgence relation obtained in Subsection 4.3 from the asymptotics of the coefficients in the perturbative series for the energy density. We then compare the resummation of the trans-series of this observable to a high precision direct solution of the Lieb–Liniger integral equation as a reference. Finally, we repeat a similar, but more elaborate analysis in terms of the physical expansion variable $g$, and also for the higher conserved charges. In the end we discuss the results in Section 8 and draw our conclusions.

## 2 Perturbative solution of the Lieb–Liniger model

In this section we review the perturbative solution of the Lieb–Liniger model based on [4]. The Hamiltonian of this model is given by

$$H = -\sum_{k=1}^{N} \frac{\partial^2}{\partial x_k^2} + 2c \sum_{1 \le j < k \le N} \delta(x_j - x_k) \,, \tag{1}$$

with a repulsive interaction $c > 0$. The system contains $N$ bosonic particles and is of size $L$ with periodic boundary conditions. Choosing an attractive interaction $c < 0$ instead would lead to an integral equation, which is equivalent to that of the Gaudin–Yang model. That model is of fermionic type, which requires a separate study. See [22] for details. We are interested in the thermodynamic limit in which the number of particles and the size become large $L, N \to \infty$, while the density $n = N/L$ is fixed. The integral equation which describes the density of Bethe roots $f(x)$ in the ground-state is given by

$$\frac{f(x)}{2} - \frac{1}{2\pi} \int_{-B}^{B} \frac{f(y)\,\mathrm{d}y}{(x-y)^2 + 1} = 1 \,. \tag{2}$$

The *physical* coupling is related to the density as $g^2 = \frac{c}{4\pi^2 n}$, which is dimensionless, and can be written as

$$\frac{1}{\pi g^2} = \int_{-B}^{B} f(x)\,\mathrm{d}x \,. \tag{3}$$

The ground state energy density of the model can be obtained from the density of Bethe roots as

$$e(g) = 16\pi^5 g^6 \int_{-B}^{B} x^2 f(x)\,\mathrm{d}x\,. \tag{4}$$

We will be interested in the moments of density of Bethe roots defined by

$$\phi_\ell = \int_{-B}^{B} x^{2\ell} f(x)\,dx\,. \tag{5}$$

The zeroth moment for $\ell = 0$ corresponds to the density, the first for $\ell = 1$ to the energy density, while higher order ones to the vacuum expectation values of higher conserved charges. We would like to calculate the small-$g$ expansion of these quantities. We will find perturbative $g^n$ and non-perturbative $e^{-1/g}$ corrections, and the complete answer will be a double expansion in these parameters: a trans-series

$$\phi_\ell = \sum_{n=0}^{\infty} e^{-n/g} \phi_\ell^{(n)} \quad ; \qquad \phi_\ell^{(n)} = \sum_{k=k_0(n,\ell)}^{\infty} \phi_{\ell,k}^{(n)} g^k\,. \tag{6}$$

where the summation starts from an $n, \ell$ dependent value. The perturbative part is $\phi_\ell^{(0)}$ and the rest contains the non-perturbative contributions.

Exact perturbative results for small coupling can be obtained by adapting [11, 12], a technique developed by Volin in [13, 14]. By applying this method, the integral equation (2) is transformed into algebraic equations which can be solved recursively. As a result, the coefficients of the perturbative series $\phi_\ell^{(0)}$ can be determined to high orders. In order to do this, the integral equation is rewritten as a difference equation for the resolvent function $R(x)$. This resolvent is related to the density of Bethe roots as

$$R(x) = \int_{-B}^{B} \frac{f(y)}{x - y}\mathrm{d}y\,, \tag{7}$$

which is analytic on the complex plane and is only discontinuous on the interval $[-B, B]$. The density of Bethe roots is then given by its jump in this interval:

$$f(x) = -\frac{1}{2\pi i}(R^+(x) - R^-(x))\,, \tag{8}$$

where $R^\pm(x) = R(x \pm i\epsilon)$. With the help of the resolvent, the integral equation (2) can be brought into the form of difference equations [11]. We would like to compute the resolvent $R(x)$ as a large-$B$ (small-$g$) expansion. To obtain this expansion, one considers the *bulk* and the *edge* regime of the resolvent, which correspond to the limits

$$\begin{aligned} \text{bulk regime:} \quad & B\,, x \to \infty \quad \text{with} \quad u = \frac{x}{B} \quad \text{fixed}\,, \\ \text{edge regime:} \quad & B\,, x \to \infty \quad \text{with} \quad z = 2(x - B) \quad \text{fixed}\,. \end{aligned} \tag{9}$$

Matching the two asymptotic expansions in the two regimes allows then to find the coefficients of the respective expansions.

**Bulk region.** The resolvent in the bulk region for the Lieb–Liniger model was proposed in [11] as[1]

$$R_B(x) = -2\pi\sqrt{x^2 - B^2} + \sum_{n=0}^{\infty} \sum_{m=0}^{\infty} \sum_{k=0}^{n+m+1} \frac{c_{n,m,k}\left(\frac{x}{B}\right)^{\epsilon(k)}}{B^{m-n-1}(x^2 - B^2)^{n+\frac{1}{2}}} \left[\log\left(\frac{x - B}{x + B}\right)\right]^k\,, \tag{10}$$

---

[1]Note that formula (10) becomes the correct resolvent only if we add a $+2\pi x$ term, see [11] for details. Then the expansion in (21) would indeed start with a $1/x$ term.

where $\epsilon(k) = k \mod 2$. At this point, we should stress that in order to capture non-perturbative contributions, the resolvent should be of a trans-series form. However, the ansatz above only captures the perturbative part. Of course, we could extend it by adding contributions that are exponentially suppressed by powers of $e^{-2\pi B}$, but the inverse Laplace transform would mix non-perturbative orders, making the analysis very complicated. Instead, we borrow results from the non-perturbative solution of the $O(3)$ model [16] and reinstate the non-perturbative corrections in Section 4. Thus in this section we calculate $\phi_\ell$ only up to the non-perturbative corrections.

The comparison between the bulk and edge regions is done for the inverse Laplace transform of the resolvent:

$$\bar{R}(s) = \frac{1}{2\pi i} \int_{-i\infty}^{i\infty} e^{sz} R\Big(B + \frac{z}{2}\Big) \mathrm{d}z. \tag{11}$$

In the bulk region, we expand (10) in powers of small $z$ and apply the inverse Laplace transform to each term of the expansion. After massaging the resulting expressions, we obtain the following result for the resolvent in the bulk region:

$$\bar{R}_B(s) = -2\pi\sqrt{B} \sum_{j=0}^{\infty} \frac{\Gamma(\frac{3}{2})}{\Gamma(j+1)\Gamma(\frac{3}{2}-j)} \frac{s^{-j-\frac{3}{2}}}{\Gamma(-j-\frac{1}{2})} (4B)^{-j}$$
$$+ \frac{\sqrt{B}}{\sqrt{s}} \sum_{m=0}^{\infty} \sum_{n=-m}^{\infty} \sum_{t=0}^{n+m+1} B^{-m} s^n [\log(4Bs)]^t V_c[n,m,t], \tag{12}$$

where

$$V_c[n,m,t] = \sum_{k=t}^{n+m+1} \sum_{j=\max(0,-n)}^{m} c_{n+j,m-j,k} F[n,t,k,j], \tag{13}$$

and

$$F[n,t,k,j] = 2^{-2j}(-1)^t \frac{\Gamma(k+1)}{\Gamma(j+1)\Gamma(t+1)\Gamma(k-t+1)}$$
$$\times \frac{\mathrm{d}^{k-t}}{\mathrm{d}x^{k-t}} \left[ \frac{\Gamma(-x-n+\frac{1}{2}-j)}{\Gamma(-x+n+\frac{1}{2})\Gamma(-x-n+\frac{1}{2}-2j)} \left(1 + \frac{2j\epsilon(k)}{-n-x+\frac{1}{2}-2j}\right) \right]_{x=0}, \tag{14}$$

which now takes a similar form as the one worked out for the $O(3)$ model in [14]. Note that the first sum in (12) corresponds to the explicitly known leading order term $-2\pi\sqrt{x^2 - B^2}$ in (10). We can bring this term to a form, that resembles the second sum in (12) and finally write the inverse Laplace transform of the resolvent in the bulk region as

$$\bar{R}_B(s) = \frac{\sqrt{B}}{\sqrt{s}} \sum_{m=0}^{\infty} \sum_{n=-m-1}^{\infty} \sum_{t=0}^{n+m+1} B^{-m} s^n [\log(4Bs)]^t$$
$$\times \sum_{k=t}^{n+m+1} \sum_{j=\max(0,-n-1)}^{m} \hat{c}_{n+j,m-j,k} F[n,t,k,j]. \tag{15}$$

Here we introduced the generalised coefficients $\hat{c}$. The first term of (12) is captured by extending the sums to include $n+j = -1$ with the coefficients $\hat{c}_{-1,0,0} = -2\pi$ and $\hat{c}_{-1,a,b} = 0$ for $a \neq 0$ or $b \neq 0$.

**Edge region.** In the edge region, the resolvent can be parametrized based on the leading order Wiener–Hopf type solution of the integral equation [11] as

$$\bar{R}(s) = \Phi_B(s)\Big(\frac{1}{s} + Q_B(s)\Big), \tag{16}$$

where the functions $\Phi_B(s)$ and $Q_B(s)$ are given by

$$
\Phi_B(s) = \frac{\sqrt{\pi B}}{\sqrt{s}} \exp\left[\frac{s}{\pi}\log\left(\frac{\pi e}{s}\right)\right]\Gamma\left(\frac{s}{\pi}+1\right),
$$

$$
Q_B(s) = \frac{1}{Bs}\sum_{m=0}^{\infty}\sum_{n=0}^{m+1}\frac{Q_{n,m+1-n}(\log B)}{B^m s^n}. \tag{17}
$$

We now need to expand this expression in $B$ and $s$ at infinity. After some work, the resolvent in the edge region can be brought into the following form:

$$
\bar{R}(s) = \frac{\sqrt{B}}{\sqrt{s}}\sum_{m=0}^{\infty}\sum_{j=-m-1}^{\infty}\sum_{i=0}^{j+m+1}\frac{s^j}{B^m}\log(4Bs)^i V_Q^{B_0}[m,j+1,i], \tag{18}
$$

where we introduced

$$
V_Q^{B_0}[m,j,i] = \sum_{k=0}^{j+m}\binom{k}{i}\log\left(\frac{B_0}{B}\right)^{k-i}V_Q[m,j,k], \tag{19}
$$

with $B_0 = \frac{1}{4\pi e}$.

**Determining the coefficients.** We can now use the recursive algorithm of Volin [13, 14] to solve for the coefficients. By comparing the expansions for the bulk and edge region, we need to solve

$$
V_c[m,n,t] = V_Q^{B_0}[m,n+1,t], \tag{20}
$$

in order to determine the respective coefficients.

## 2.1 Extracting the moments

From the definition of the resolvent $R(x)$ in (7), we can see how it is related to the sought for moments (5)

$$
R_B(x) = \int_{-\infty}^{\infty}\sum_{\ell\geq 0}\frac{f(y)y^{2\ell}}{x^{2\ell+1}}\mathrm{d}y = \sum_{\ell\geq 0}\phi_\ell x^{-2\ell-1}. \tag{21}
$$

The moments can be obtained by expanding the resolvent in the bulk region from eq. (10) at $x=\infty$. For instance, from the large $x$ expansion we can easily read off the first few moments:

$$
\phi_0 = \rho = \pi B^2 + \sum_{m=0}^{\infty}\frac{c_{0,m,0}-2c_{0,m,1}}{B^{m-1}},
$$

$$
\phi_1 = \frac{\pi B^4}{4} + \sum_{m=0}^{\infty}\left(\frac{c_{1,m,0}-2c_{1,m,1}}{B^{m-2}} + \frac{\frac{1}{2}c_{0,m,0}-\frac{5}{3}c_{0,m,1}+4c_{0,m,2}-8c_{0,m,3}}{B^{m-3}}\right).
$$

$$
\phi_2 = \frac{\pi B^5}{8} + \sum_{m=0}^{\infty}\left(\frac{c_{2,m,0}-2c_{2,m,1}}{B^{m-2}} + \frac{\frac{3}{2}c_{1,m,0}-\frac{11}{3}c_{1,m,1}+4c_{1,m,2}-8c_{1,m,3}}{B^{m-3}}\right.
$$

$$
\left. + \frac{\frac{3}{8}c_{0,m,0}-\frac{89}{60}c_{0,m,1}+\frac{14}{3}c_{0,m,2}-12c_{0,m,3}}{B^{m-5}}\right). \tag{22}
$$

Other moments can be obtained by expanding $R_B(x)$ to higher orders. Using the recursive algorithm, we can determine the coefficients $c_{n,m,k}$ and evaluate the perturbative part of the moments up to any order in $1/B$. However, the number of coefficients contributing to a given

order in $x$ grows quickly and it becomes laborious to evaluate higher moments. For instance, for first moments $\phi_0$ and $\phi_1$, we find

$$\phi_0 = \pi B^2 + \log\left(\frac{16\pi B}{e}\right)B + \frac{\log(16\pi B)^2 - 2}{4\pi} - \frac{1 - 2\log(16\pi B)^2 + 3\zeta_3}{16\pi^2 B} + O(B^{-2}), \quad (23)$$

$$\phi_1 = \frac{\pi B^4}{4} + \frac{-7 + 3\log(16\pi B)}{6}B^3 + \left(\frac{3(-4 + \log(16\pi B))\log(16\pi B)}{8\pi} + \frac{\pi}{6}\right)B^2 + O(B).$$

These expressions contain $\log(B)$ terms and the coefficients to higher orders become rather bulky. In the analysis of the $O(N)$ sigma models it was found [17], that by introducing a running coupling $v$, terms with $\log B$ can be resummed. This coupling is defined through

$$2\pi B = \frac{1}{v} + \log\left(\frac{v}{8e}\right). \quad (24)$$

Substituting this in the results for the moments, the expressions take a much nicer form. For example, the first 5 perturbative coefficients of the moments are given by

$$\phi_0 = \frac{1}{\pi}\left[\frac{1}{4v^2} - \frac{1}{v} - \frac{1}{4} + \frac{1 - 3\zeta_3}{8}v + \frac{5 - 33\zeta_3}{24}v^2 + O(v^3)\right],$$

$$\phi_1 = \frac{1}{\pi^3}\left[\frac{1}{64v^4} - \frac{5}{24v^3} + \frac{45 + 4\pi^2}{96v^2} + \frac{63\zeta_3 - 32\pi^2 + 51}{192v} + \frac{-9\zeta_3 - 8\pi^2 - 16}{192} + O(v^1)\right],$$

$$\phi_2 = \frac{1}{\pi^5}\left[\frac{1}{512v^6} - \frac{89}{1920v^5} + \frac{439 + 24\pi^2}{1536v^4} + \frac{981\zeta_3 - 640\pi^2 - 1207}{3072v^3}\right.$$
$$\left. + \frac{-41775\zeta_3 + 576\pi^4 + 7200\pi^2 - 6395}{15360v^2} + O(v^{-1})\right],$$

$$\phi_3 = \frac{1}{\pi^7}\left[\frac{5}{16384v^8} - \frac{381}{35840v^7} + \frac{6949 + 300\pi^2}{61440v^6} + \frac{-53253 - 14240\pi^2 + 19575\zeta_3}{122880v^5}\right. \quad (25)$$
$$\left. + \frac{41414 + 87800\pi^2 + 4320\pi^4 - 360897\zeta_3}{122880v^4} + O(v^{-3})\right],$$

$$\phi_4 = \frac{1}{\pi^9}\left[\frac{7}{131072v^{10}} - \frac{25609}{10321920v^9}\right.$$
$$+ \frac{317027 + 11760\pi^2}{8257536v^8} + \frac{5258925\zeta_3 - 4096512\pi^2 - 20887535}{82575360v^7}$$
$$\left. + \frac{-153629307\zeta_3 + 1693440\pi^4 + 43584128\pi^2 + 54842251}{82575360v^6} + O(v^{-5})\right].$$

## 3 Differential relations

In the derivations of the next sections we will rely heavily on a method that was developed [18, 21, 22] to solve the free-energy problem of certain relativistic models, where a similar type of integral equation appears.[2] The ODEs we present in this section were initially developed in [7, 23] to relate the moments $\phi_\ell$ with each other, however, here we will need to combine them also with their generalizations [18, 21]. Our definitions here may seem ad-hoc, yet the objects we introduce here will be useful in deriving the full Wiener–Hopf solution to the original problem in Section 4.

---

[2]Most importantly, among these models is the $O(3)$ symmetric non-linear sigma model [17, 21], which has the same kernel function.

## 3.1 Generalized moments

We extend the integral equation (2) by introducing a more general source term:

$$\chi_\alpha(\theta) - \int_{-b}^{b} K(\theta - \theta')\chi_\alpha(\theta')\mathrm{d}\theta' = \cosh(\alpha x), \quad |\theta| \leq b, \tag{26}$$

where $\alpha \geq 0$ is a real parameter and $1/K(\theta) = \theta^2 + \pi^2$. In particular, $\alpha = 0$ recovers the original integral equation if we set

$$b = \pi B, \qquad f(x) = 2\chi_0(\pi x). \tag{27}$$

We can now consider the densities

$$\mathcal{O}_{\alpha,\beta} \equiv \frac{1}{2\pi} \int_{-b}^{b} \cosh(\beta\theta)\chi_\alpha(\theta)\mathrm{d}\theta, \tag{28}$$

with $\beta \geq 0$, which are symmetric in $\alpha$ and $\beta$. In the $\alpha = \beta = 0$ case, we recover the density $\phi_0 = 4\mathcal{O}_{0,0}$. For generic parameters, $\mathcal{O}_{\alpha,\beta}$ can be used to generate the *generalized moments* of the solution $\chi_\alpha(\theta)$:

$$\phi_{\alpha;\ell} = \frac{4}{\pi^{2\ell}}\frac{1}{2\pi}\int_{-b}^{b}\chi_\alpha(\theta)\theta^{2\ell}\mathrm{d}\theta = \frac{4}{\pi^{2\ell}}\partial_\beta^{2\ell}\mathcal{O}_{\alpha,\beta}\Big|_{\beta=0}. \tag{29}$$

For $\alpha = 0$ they are the usual moments $\phi_\ell = \phi_{0;\ell}$. It will also be useful to consider the values of $\chi_\alpha(\theta)$ at the boundaries of the integration range, which we denote as

$$\chi_\alpha \equiv \chi_\alpha(\pm b). \tag{30}$$

From now on, we use a dot to denote the total derivative with respect to $b$.

Straightforward manipulations (see Appendix A of [21] for the detailed derivations) lead to the ODEs relating the $\mathcal{O}_{\alpha,\beta}$-s and $\chi_\alpha$-s:

$$\dot{\mathcal{O}}_{\alpha,\beta} = \frac{1}{\pi}\chi_\alpha\chi_\beta \tag{31}$$

$$\ddot{\mathcal{O}}_{\alpha,\beta} - 2\frac{\dot{\chi}_\alpha}{\chi_\alpha}\dot{\mathcal{O}}_{\alpha,\beta} + (\alpha^2 - \beta^2)\mathcal{O}_{\alpha,\beta} = 0 \tag{32}$$

$$\pi(\alpha^2 - \beta^2)\mathcal{O}_{\alpha,\beta} = \dot{\chi}_\alpha\chi_\beta - \chi_\alpha\dot{\chi}_\beta \tag{33}$$

$$\frac{\ddot{\chi}_\alpha}{\chi_\alpha} - \alpha^2 = F. \tag{34}$$

In the last line, $F \equiv \ddot{\chi}_0/\chi_0$ is an universal, $\alpha$-independent function of $b$. Not all of these equations are independent: the second-order differential equations (32), (34) can be derived from the first-order ones (31), (33).

From (32), we may obtain an expression relating the moment $\phi_\ell$ with the previous moment $\phi_{\ell-1}$. Taking $2\ell$ derivatives with respect to $\beta$ in (32), then evaluating at $\alpha = \beta = 0$ and using the relation (29), we arrive at [7]

$$\ddot{\phi}_\ell - \frac{\dddot{\phi}_0}{\ddot{\phi}_0}\dot{\phi}_\ell = 2\ell(2\ell - 1)\frac{\phi_{\ell-1}}{\pi^2}, \tag{35}$$

where we used (31) to rewrite $2\dot{\chi}_0/\chi_0 = \dddot{\phi}_0/\ddot{\phi}_0$. This equation can be used recursively to compute $\phi_\ell$ from $\phi_{\ell-1}$, with initial condition given by $\phi_0$.

The generalized moments $\phi_{\alpha;\ell}$ in (29) may be obtained directly from the moments at $\alpha = 0$. Indeed, if we take $2\ell$ derivatives with respect to $\beta$ in (31), both at generic $\alpha$ and at $\alpha = 0$, we can then combine both equations into the relation

$$\dot{\phi}_{\alpha;\ell} = \frac{\chi_\alpha}{\chi_0}\dot{\phi}_\ell. \tag{36}$$

All the above equations may be used to determine the generalized moments $\phi_{\alpha;\ell}$ (up to some integration constants) by taking the number density $\rho = \phi_0$ as a starting point. Schematically, the following steps must be taken to compute $\phi_{\alpha;\ell}$:

$$
\begin{array}{ccc}
\phi_0 & \xrightarrow{(35)} & \phi_\ell \\[4pt]
{\scriptstyle(31)}\downarrow & & \vdash\!\!\xrightarrow{(36)}\;\phi_{\alpha;\ell} \\[4pt]
\chi_0 & \xrightarrow[(34)]{} & \chi_\alpha
\end{array}
$$

We will use the above procedure to determine only the perturbative parts of these objects, rather than their full trans-series. To obtain the full trans-series of the $\phi_\ell$ moments, for the non-perturbative corrections we will also need the perturbative parts of the $\mathcal{O}_{\alpha,\beta}$-s as well (these can be calculated for example via (31) from $\chi_\alpha$). Clearly, the perturbative parts of the quantities $\mathcal{O}_{\alpha,\beta}$, $\chi_\alpha$, $\phi_\ell$ satisfy the same differential equations as their full expressions.

All of the differential relations are simpler to solve (although longer to express) in terms of the running coupling $v$, where after proper normalization (see Section 4) each of the above objects' perturbative parts are simply (asymptotic) power serieses of $v$.

Rewriting (35) in terms of $\rho$-derivatives, it simplifies as

$$
\frac{\mathrm{d}^2\phi_\ell}{\mathrm{d}\rho^2} = 2\ell(2\ell-1)\frac{\phi_{\ell-1}}{(2\chi_0)^4}. \tag{37}
$$

Thus the structure of the solution must be the following:

$$
\phi_\ell = \hat{\phi}_\ell + p_\ell\rho + q_\ell, \tag{38}
$$

where $\hat{\phi}_\ell$ is a particular solution to (37), and we need to fix the constants of the homogeneous solution $p_\ell, q_\ell$ from elsewhere. We will turn to this problem in Subsection 3.2, however, there we will use the variable $v$ instead of $\rho$.

The works [7, 23] rewrite (35) in terms of the dimensionless parameter $\gamma = c/n = (2\pi g)^2 = 4\pi/\rho$ to generate the higher moments

$$
e_{2\ell} \equiv \gamma^{2\ell+1}\frac{\phi_\ell}{4\pi}, \quad \ell > 1 \tag{39}
$$

from existing data for the energy density $e = e_2$ in the following way:

$$
\frac{\mathrm{d}^2}{\mathrm{d}\gamma^2}\left(\frac{e_{2\ell}}{\gamma^{2\ell}}\right) = \ell(2\ell-1)\left[\frac{\mathrm{d}^2}{\mathrm{d}\gamma^2}\left(\frac{e_2}{\gamma^2}\right)\right]\left(\frac{e_{2(\ell-1)}}{\gamma^{2(\ell-1)}}\right) \quad \text{for} \quad \ell > 1. \tag{40}
$$

These relations should provide us a way to study the resurgence relations of the higher moments directly in the physical coupling based on the trans-series of $e$. However, the latter itself has to be calculated from

$$
\frac{\mathrm{d}^2}{\mathrm{d}\gamma^2}\left(\frac{e_2}{\gamma^2}\right) = \frac{2\pi^2}{(\gamma\chi_0)^4}, \tag{41}
$$

i.e. (37) taken at $\ell = 1$. As we will see, the trans-series structure of $\chi_0$ is only known in terms of the running coupling $v$, and we did not find a way to express it in terms of $\gamma$ or $g$ explicitly. Thus, in the next sections we will determine the resurgent properties of $\phi_\ell$ in terms of $v$ instead, using the $\phi_{\alpha;\ell}$ and $\mathcal{O}_{\alpha,\beta}$ generalized moments. Then we rewrite it in terms of the physical coupling $g$ in a direct way (while truncating the trans-series at finite orders).

## 3.2  Integration constants

Once we compute the zeroth moment $\phi_0$ from Volin's recursive algorithm described in Section 2, we would like to obtain higher moments $\phi_\ell$ using the differential equation (35). As was already mentioned, a solution $\phi_\ell$ to this differential equation has two free parameters and in this section we will discuss how to fix them.

We recall that the differential equation (35) is written in terms of derivatives with respect to $B$, and we would like to rewrite this expression in terms of $v$ derivatives. From the definition of the $v$ coupling in (24), we easily find

$$\frac{\mathrm{d}v}{\mathrm{d}b} = -\frac{2v^2}{1-v}. \tag{42}$$

The chain rule then leads to the following equivalent differential equation:

$$\frac{4v^4}{(1-v)^2}\left(\phi_\ell'' - \frac{\phi_0''}{\phi_0'}\phi_\ell'\right) = 2\ell(2\ell-1)\frac{\phi_{\ell-1}}{\pi^2}, \tag{43}$$

where prime denotes derivative with respect to $v$.

We assume the following perturbative ansatz for the moments[3]

$$\phi_\ell^{(0)} = \frac{1}{\pi^{2\ell+1}}\frac{1}{v^{2\ell+2}}\sum_{k\geq 0}\varphi_{\ell,k}^{(0)}v^k. \tag{44}$$

We cannot prove that $\phi_\ell^{(0)}$ has a regular power series expansion in $v$ without $\log v$ terms, but this is compatible with all the terms we calculated from Volin's method and justified a posteriori by comparing to the numerical solution of the problem.

The leading behaviour in $v$ can be deduced e.g. from expanding the leading order term $2\pi\left(x - \sqrt{x^2 - B^2}\right)$ of the bulk ansatz for large $x$, see also the footnote at (10). When plugging this expression in (43), we obtain the following constraint between the perturbative coefficients:

$$\varphi_{\ell,k}^{(0)} = \frac{1}{2(2\ell-k)(2\ell+2-k)\varphi_{0,0}^{(0)}}\left[\sum_{j=1}^{k}(j-2)(2\ell+2+j-k)(2\ell+2j-k)\varphi_{0,j}^{(0)}\varphi_{\ell,k-j}^{(0)}\right.$$
$$-\frac{\ell(2\ell-1)}{2}\left(\sum_{j=0}^{k}(k-j-2)\varphi_{\ell-1,j}^{(0)}\varphi_{0,k-j}^{(0)} - 2\sum_{j=0}^{k-1}(k-j-3)\varphi_{\ell-1,j}^{(0)}\varphi_{0,k-1-j}^{(0)}\right.$$
$$\left.\left.+\sum_{j=0}^{k-2}(k-j-4)\varphi_{\ell-1,j}^{(0)}\varphi_{0,k-2-j}^{(0)}\right)\right]. \tag{45}$$

Note that this result is only valid under the condition $k \neq 2\ell$ and $k \neq 2\ell + 2$, which corresponds to the coefficients of $1/v^2$ and $v^0$ in the perturbative expansion of $\phi_\ell^{(0)}$. These two coefficients are precisely the two integration constants for solutions to the differential equation (43). One way to partially address this issue, as discussed in [7], is to consider (45) with $\ell \mapsto \ell + 1$ and

---

[3]We denote the perturbative coefficients of $\phi_\ell^{(0)}$ in the coupling $v$ as $\varphi_{\ell,k}^{(0)}$ to avoid confusion with the notation introduced in (6) for the coefficients of the same quantity in the coupling $g$.

solve for the coefficient $\varphi_{\ell,2\ell+2}^{(0)}$ in the resulting expression:

$$
\begin{aligned}
\varphi_{\ell,2\ell+2}^{(0)} = -\frac{1}{(\ell+1)(2\ell+1)\varphi_{0,0}^{(0)}} \Bigg[ &\sum_{j=1}^{2\ell+2} (j-2)(j+2)(2j)\varphi_{0,j}^{(0)}\varphi_{\ell+1,2\ell+2-j}^{(0)} \\
-\frac{(\ell+1)(2\ell+1)}{2}\Bigg( &\sum_{j=0}^{2\ell+1}(2\ell-j)\varphi_{\ell,j}^{(0)}\varphi_{0,2\ell+2-j}^{(0)} - 2\sum_{j=0}^{2\ell+1}(2\ell-j-1)\varphi_{\ell,j}^{(0)}\varphi_{0,2\ell+1-j}^{(0)} \\
&+\sum_{j=0}^{2\ell}(2\ell-j-2)\varphi_{\ell,j}^{(0)}\varphi_{0,2\ell-j}^{(0)}\Bigg)\Bigg].
\end{aligned}
\tag{46}
$$

Note that we compute $\varphi_{\ell,2\ell+2}^{(0)}$ using the higher moment coefficients $\varphi_{\ell+1,j}^{(0)}$, $0 \leq j \leq 2\ell+1$, $j \neq 2\ell$, which can be obtained from (45).

We are still left with one undetermined coefficient, $\varphi_{\ell,2\ell}^{(0)}$, in each moment. We have to fix this remaining coefficient by directly computing the moment $\phi_\ell$ up to order $1/v^2$ with Volin's recursive algorithm, discussed in Section 2. It is then possible to compute $\phi_\ell$ up to any perturbative order in $v$ using (45) and (46).

## 4  Wiener–Hopf solution

The integral equation (26) can be solved by the Wiener–Hopf technique [24, 25, 9, 15, 17, 22]. The idea is to extend the integral equation for the whole line by introducing an unknown function, non-vanishing only outside the interval $[-b, b]$. Using Fourier transformation the kernel can be easily inverted. The introduced unknown function and $f(x)$ can be separated by the different analytical behaviours of their Fourier transform. This requires the following Wiener–Hopf factorisation:

$$
\frac{1}{1-\tilde{K}(\omega)} = G_+(\omega)G_-(\omega),
\tag{47}
$$

where $\hat{K}(\omega)$ is the Fourier transform of the kernel $K(\theta)$ in (26), $G_+(\omega)$ is an analytic function on the upper half plane, while $G_-(\omega) = G_+(-\omega)$ is analytic on the lower half plane. Explicitly we have

$$
G_+(i\kappa) = \frac{\Gamma\left(1+\frac{\kappa}{2}\right)}{\sqrt{\pi\kappa}}e^{-\frac{\kappa}{2}(\ln\kappa-1-\ln 2)} = \frac{1}{\sqrt{\pi\kappa}} + \sqrt{\kappa}(a_0 + a_1\log\kappa) + \ldots,
\tag{48}
$$

where we have also presented the structure of the leading terms in its small $\kappa$ expansion. The full trans-series solution of the various observables was obtained in [18] in terms of a perturbatively defined basis $A_{\alpha,\beta}$. This basis is related to the perturbative part of the observable $\mathcal{O}_{\alpha,\beta}$ as

$$
\mathcal{O}_{\alpha,\beta}^{(0)} = \frac{1}{4\pi}G_+(i\alpha)G_+(i\beta)e^{(\alpha+\beta)b}A_{\alpha,\beta}.
\tag{49}
$$

together with the perturbative parts of $\chi_\alpha$

$$
\chi_\alpha^{(0)} = \frac{1}{2}G_+(i\alpha)e^{\alpha b}a_\alpha,
\tag{50}
$$

valid for $\alpha \neq 0$, $\beta \neq 0$. Otherwise, the normalization constants are

$$
\mathcal{O}_{0,\beta}^{(0)} = \frac{1}{2\pi}G_+(i\beta)e^{\beta b}A_{0,\beta} \quad ; \quad \mathcal{O}^{(0)} = \frac{1}{\pi}A_{0,0} \quad ; \quad \chi_0^{(0)} = a_0.
\tag{51}
$$

It can be shown from (31) and (34) that they satisfy the following system of differential equations:

$$
(\alpha+\beta)A_{\alpha,\beta} + \dot{A}_{\alpha,\beta} = a_\alpha a_\beta \quad ; \quad \ddot{a}_\alpha + 2\alpha\dot{a}_\alpha = fa_\alpha,
\tag{52}
$$

for all $\alpha, \beta$, where $f$ is the perturbative part of $F$. These differential equations can be used to calculate $a_0$ and $f$ from the already determined $A_{0,0}$, uniquely. Solving the differential equations for generic $\alpha$, $\beta$ we also obtain a unique perturbative solution. The solutions obtained are meaningful for negative $\alpha$ and $\beta$ and can be used to express the full non-perturbative solutions. The two quantities $A_{\alpha,\beta}$ and $a_\alpha$ are related to each other as $\lim_{\beta \to \infty} \beta A_{\alpha,\beta} = a_\alpha$. Since $G_+(0)$ is infinite, we need to use a different normalization for $A_{0,\beta}, A_{0,0}$ and $a_0$.

The densities $\mathcal{O}_{\alpha,\beta}$ defined in (28) can be written as a trans-series in terms of the perturbative quantities $A_{\alpha,\beta}$. For $\alpha = 0$, $\beta \neq 0$, we have

$$\mathcal{O}_{0,\beta} = \frac{1}{2\pi}\left[ G_+(i\beta)e^{\beta b}\hat{A}_{0,\beta} + G_-(i\beta)e^{-\beta b}\hat{A}_{0,-\beta} \right]. \tag{53}$$

Here and throughout the paper, as an abuse of notation, whenever we write $G_-(i\kappa)$ for $\kappa > 0$ we mean the limit $G_-(i\kappa + 0)$ that has to be carefully evaluated due to the cut along the positive imaginary line in $G_-(\omega)$. For $\alpha, \beta \neq 0$, instead we have

$$\mathcal{O}_{\alpha,\beta} = \frac{1}{4\pi}\left[ G_+(i\alpha)G_+(i\beta)e^{(\alpha+\beta)b}\hat{A}_{\alpha,\beta} + G_-(i\alpha)G_+(i\beta)e^{(-\alpha+\beta)b}\hat{A}_{-\alpha,\beta} \right.$$
$$\left. + G_+(i\alpha)G_-(i\beta)e^{(\alpha-\beta)b}\hat{A}_{\alpha,-\beta} + G_-(i\alpha)G_-(i\beta)e^{-(\alpha+\beta)b}\hat{A}_{-\alpha,-\beta} \right], \tag{54}$$

where $\hat{A}_{\alpha,\beta}$ can be written as a trans-series in terms of the quantities $A_{\alpha,\beta}$:

$$\hat{A}_{\alpha,\beta} = A_{\alpha,\beta} + \sum_{r,s=1}^{\infty} A_{\alpha,-\kappa_r}\mathcal{A}_{-\kappa_r,-\kappa_s}A_{-\kappa_s,\beta},$$

$$\mathcal{A}_{-\kappa_r,-\kappa_s} = \sum_{N=-1}^{\infty}\sum_{j_1,j_2,\ldots,j_N=1}^{\infty} S_{\kappa_r}e^{-2\kappa_r b}A_{-\kappa_r,-\kappa_{j_1}}S_{\kappa_{j_1}}e^{-2\kappa_{j_1}b}A_{-\kappa_{j_1},-\kappa_{j_2}} \tag{55}$$
$$\cdots S_{\kappa_{j_N}}e^{-2\kappa_{j_N}b}A_{-\kappa_{j_N},-\kappa_s}S_{\kappa_s}e^{-2\kappa_s b}.$$

Here the $N = -1$ term is formally to be understood as $S_{\kappa_r}e^{-2\kappa_r b}\delta_{-\kappa_r,-\kappa_s}$, while the $N = 0$ term is simply $S_{\kappa_r}e^{-2\kappa_r b}A_{-\kappa_r,-\kappa_s}S_{\kappa_s}e^{-2\kappa_s b}$. This expression involves the poles of the function $\sigma(\omega) = G_-(\omega)/G_+(\omega)$, which are located at $\kappa_r = 2r$, $r \in \mathbb{Z}$, $r \geq 1$, with residues

$$S_{\kappa_r} = i\frac{2}{(r-1)!\,r!}\left(\frac{r}{e}\right)^{2r}. \tag{56}$$

evaluated again at $i\kappa_r + 0$.

Similarly, we might write a trans-series for the boundary values in (30):

$$\chi_\alpha = \frac{1}{2}G_+(i\alpha)e^{\alpha b}\hat{a}_n + \frac{1}{2}G_-(i\alpha)e^{-\alpha b}\hat{a}_{-\alpha}, \tag{57}$$

where

$$\hat{a}_\alpha = a_\alpha + \sum_{r,s=1}^{\infty} A_{\alpha,-\kappa_r}\mathcal{A}_{-\kappa_r,-\kappa_s}a_{-\kappa_s}. \tag{58}$$

## 4.1 Exponential corrections to the moments

We would like to compute exponential corrections to the moments $\phi_\ell$ in a similar way to how the trans-series of the densities are computed in (53), (54) and (55). Specifically, we would like to take derivatives with respect to $\beta$ in (54) to obtain the trans-series for the moments. It is

instructive to collect the appearance of the $\beta$ dependence of the various observables. They all contain the combination

$$\frac{1}{2}\left[G_+(i\beta)e^{\beta b}A_{\alpha,\beta} + G_-(i\beta)e^{-\beta b}A_{\alpha,-\beta}\right]. \tag{59}$$

We thus define the perturbative building blocks of the non-relativistic moments $A_{\alpha;\ell}$ from its derivatives

$$A_{\alpha;\ell} = \lim_{\beta\to 0} \partial_\beta^{2\ell} \frac{1}{2}\left[G_+(i\beta)e^{\beta b}A_{\alpha,\beta} + G_-(i\beta)e^{-\beta b}A_{\alpha,-\beta}\right]. \tag{60}$$

We will not need the explicit form of these building blocks, but we need to see that they are well defined and the limit exists. The expression is particularly worrysome as at small $\beta$ we have $G_+(\pm i\beta) \sim 1/\sqrt{\beta}$. Additionally, the $\beta \to 0$ limit is not well-defined in terms of the perturbative definition of $A_{\alpha,\beta}$. This is because it is a large $b$ expansion, and this asymptotic series is singular for $\alpha, \beta \to 0$ (see e.g. Appendix C of [22]). In order to calculate the correct small $\beta$ limit of $A_{\alpha,\beta}$ we have to exploit that it satisfies the Wiener–Hopf integral equation at the perturbative level, see for instance (57,59) in [22]. The small $\beta$ behaviour then can be calculated following (2.16-2.19) of [17], which results in the form $A_{\alpha,\beta} \sim \sqrt{\beta}(b_0 + b_1 \log(\beta) + \dots$. Then one can explicitely show that in the $\beta \to 0$ limit of (59) the square-root type singularities of $G_\pm(i\beta)$ together with the logarithms cancel, and the combination has a well-defined limit. We circumvent the explicit calculations by leveraging that the limit exists, and providing a perturbative definition of $A_{\alpha;\ell}$ based on the ODEs (35) and (36).

In particular, we note that we can construct $A_{\alpha;\ell}$ directly from the leading exponential correction in $\phi_{\alpha;\ell}$:

$$\phi_{\alpha;\ell} = \frac{4}{\pi^{2\ell+1}} \cdot \frac{1}{2}G_+(i\alpha)e^{\alpha b}\left[A_{\alpha;\ell} + O\left(e^{-\min(2\alpha,4)b}\right)\right]. \tag{61}$$

From the differential equation (36), we can derive a new differential equation that relates the perturbative objects $A_{\alpha;\ell}$:

$$\dot{A}_{\alpha;\ell} + \alpha A_{\alpha;\ell} = \frac{a_\alpha}{a_0}\dot{A}_{0;\ell}, \tag{62}$$

where $a_\alpha$ was defined in (57) and (58). This equation provides a method to compute exponential corrections to the moment $\phi_\ell$ from its perturbative part, $\phi_\ell^{(0)} = \frac{4}{\pi^{2\ell+1}}A_{0;\ell}$.

Using then the relationship (29) between the densities $\mathcal{O}_{\alpha,\beta}$ and the moments, and applying it to (53), we obtain

$$\phi_\ell = \frac{4}{\pi^{2\ell+1}}\left[A_{0;\ell} + \sum_{r,s=1}^{\infty} A_{0,-\kappa_r}\mathcal{A}_{-\kappa_r,-\kappa_s}A_{-\kappa_s;\ell}\right]. \tag{63}$$

Observe that the perturbative part is simply $\phi_\ell^{(0)} = \frac{4}{\pi^{2\ell+1}}A_{0;\ell}$, which we have already calculated. Using (29) again, but now applying it to the densities with $\alpha \neq 0$ in (54), we might write the analogue of (63) for the generalized moments:

$$\phi_{\alpha;\ell} = \frac{4}{\pi^{2\ell+1}}\left[\frac{1}{2}\left(G_+(i\alpha)e^{\alpha b}A_{\alpha;\ell} + G_-(i\alpha)e^{-\alpha b}A_{-\alpha;\ell}\right)\right.$$
$$\left. + \sum_{r,s=1}^{\infty}\frac{1}{2}\left(G_+(i\alpha)e^{\alpha b}A_{\alpha,-\kappa_r} + G_-(i\alpha)e^{-\alpha b}A_{-\alpha,-\kappa_r}\right)\mathcal{A}_{-\kappa_r,-\kappa_s}A_{-\kappa_s;\ell}\right]. \tag{64}$$

The differential equation (62) can be rewritten in terms of $v$ derivatives. Using the chain rule with (42), we find the following equivalent differential equation:

$$A'_{\alpha;\ell} - \frac{1-v}{2v^2}\alpha A_{\alpha;\ell} = \frac{a_\alpha}{a_0}A'_{0;\ell}. \tag{65}$$

With the perturbative ansatz

$$A_{\alpha;\ell} = \frac{1}{v^{2\ell+1/2}} \sum_{k \geq 0} A_{\alpha;\ell,k} v^k, \tag{66}$$

the coefficients that solve (65) are recursively given by

$$A_{\alpha;\ell,k} = \frac{2}{\alpha} \left[ \left( \frac{\alpha}{2} + k - 2\ell - \frac{3}{2} \right) A_{\alpha;\ell,k-1} - h_{\ell,\alpha,k} \right], \tag{67}$$

where $h_{\ell,\alpha,k}$ are the coefficients defined as

$$\frac{a_\alpha}{a_0} A'_{0;\ell} = \frac{1}{v^{2\ell+5/2}} \sum_{k \geq 0} h_{\ell,\alpha,k} v^k, \tag{68}$$

and they can be recursively defined as

$$h_{\ell,\alpha,k} = \frac{1}{a_{0,0}} \left[ \sum_{j=0}^{k} (k - j - 2\ell - 2) a_{\alpha,j} \varphi^{(0)}_{\ell,k-j} - \sum_{j=1}^{k} a_{0,j} h_{\ell,\alpha,k-j} \right]. \tag{69}$$

The coefficients of $a_\alpha$ are

$$a_0 = \frac{1}{v^{1/2}} \sum_{k \geq 0} a_{0,k} v^k, \qquad a_\alpha = \sum_{k \geq 0} a_{\alpha,k} v^k, \quad \alpha \neq 0. \tag{70}$$

With the perturbative coefficients of $A_{\alpha;\ell}$, we can now construct any exponential correction of the moment $\phi_\ell$ using (63). For example, the trans-series of the first two moments are given by

$$
\begin{aligned}
\phi_0 = \frac{1}{\pi} \Bigg[ &\frac{1}{4v^2} - \frac{1}{v} - \frac{1}{4} + \frac{1 - 3\zeta_3}{8} v + \frac{5 - 33\zeta_3}{24} v^2 + O(v^3) \\
&+ \left( \frac{32i}{v^3} - \frac{8i}{v^2} - \frac{15i}{v} - i\frac{211 - 288\zeta_3}{12} - i\frac{5695 - 33216\zeta_3}{192} v + O(v^2) \right) e^{-2/v} \\
&+ 64 \left( \frac{16 + 64i}{v^5} - \frac{8 + 40i}{v^4} - \frac{24 + 91i}{2v^3} + \frac{-616 - 2419i + (576 + 2304i)\zeta_3}{48v^2} \right. \\
&\left. + \frac{-30400 - 120543i + (148992 + 572928i)\zeta_3}{1536v} + O(v^0) \right) e^{-4/v} + O\left(e^{-6/v}\right) \Bigg]. \tag{71}
\end{aligned}
$$

$$
\begin{aligned}
\phi_1 = \frac{1}{\pi^3} \Bigg[ &\frac{1}{64v^4} - \frac{5}{24v^3} + \frac{45 + 4\pi^2}{96v^2} + \frac{63\zeta_3 - 32\pi^2 + 51}{192v} + \frac{-9\zeta_3 - 8\pi^2 - 16}{192} + O(v^1) \\
&+ \left( \frac{4i}{v^5} - \frac{25i}{v^4} + i\frac{128\pi^2 - 285}{8v^3} - i\frac{-2016\zeta_3 + 128\pi^2 - 581}{96v^2} \right. \\
&\left. - i\frac{-29376\zeta_3 + 3840\pi^2 - 27745}{1536v} + O(v^0) \right) e^{-2/v} \\
&+ 64 \left( \frac{2 + 8i}{v^7} - \frac{13 + 61i}{v^6} + \frac{(-168 + 159i) + (128 + 512i)\pi^2}{48v^5} \right. \\
&\left. + \frac{(4032 + 16128i)\zeta_3 - (512 + 2560i)\pi^2 + (3032 + 18629i)}{384v^4} \right) e^{-4/v} + O\left(e^{-6/v}\right) \Bigg]. \tag{72}
\end{aligned}
$$

## 4.2 Checking the trans-series solution

In this subsection we argue that, if the perturbative objects $A_{\alpha,\beta}$ and $a_\alpha$ satisfy the the differential equations in (52), then the trans-series (54), (57) of $\mathcal{O}_{\alpha,\beta}$, $\chi_\alpha$, constructed from these objects, are a solution to the differential equations (31) and (34). In fact, since the differential equations are not independent it is sufficient to check only (33) and (34). Details on these checks are given in the Appendices A.2 and A.3, respectively. For the latter one, we also construct the form of the non-perturbative corrections to $F$.

Similarly, we aim to prove that the trans-series of $\phi_\ell$ constructed in (63) satisfies the differential equation (35) for any $\ell$. For the interested reader, we present the technical details of this check in Appendix A.4.

For these checks, we substitute the trans-series solutions into the respective equation. Further, we assume that the perturbative versions of these equations hold. Since the non-perturbative corrections are build from the perturbative basis, we can show, that the trans-series ansatz indeed solves the equations (31)–(35).

## 4.3 Alien derivatives and median resummation

In Section 5 of [18], the cancellation of imaginary ambiguities in the resummed trans-series for $\mathcal{O}_{\alpha,\beta}$ suggested a formula for the so-called alien derivative[4] of the perturbative part $A_{\alpha,\beta}$:

$$\Delta_{\kappa_j} A_{\alpha,\beta} = 2S_{\kappa_j} A_{\alpha,-\kappa_j} A_{-\kappa_j,\beta}. \tag{74}$$

This formula was backed by numerics, and also through the work in Section 9 of [21]. Applying this result to (60), after commuting $\Delta_{\kappa_j}$ with the $\beta \to 0$ limit and the derivative, we arrive at

$$\Delta_{\kappa_j} A_{\alpha;\ell} = 2S_{\kappa_j} A_{\alpha,-\kappa_j} A_{-\kappa_j;\ell}. \tag{75}$$

We will numerically verify this formula in Section 7.

In a similar fashion as it was done in [18], it can be shown that the Stokes automorphism

$$\mathfrak{S} = \exp\left( \sum_{j=1}^{\infty} e^{-2b\kappa_j} \Delta_{\kappa_j} \right) \tag{76}$$

is able to generate the full[5] trans-series of $\phi_\ell$ (63) from its perturbative part only:[6]

$$\mathfrak{S}^{1/2} A_{0;\ell} = \phi_\ell. \tag{77}$$

Since we can only determine $A_{\alpha,\beta}$ and $A_{\alpha;\ell}$ (the building blocks of the trans-series of $\phi_\ell$) in terms of asymptotic series, whether the trans-series of $\phi_\ell$ indeed reconstructs the correct function is a remaining question. In Section 7, we test the difference between the lateral Borel resummation ($S^+$) of the trans-series (63) against a high-precision numerical solution to the

---

[4]Here by the notation $\Delta_\omega$ we mean the alien derivative of an asymptotic series in terms of integer powers of the coupling $v$, together with a slight modification compared to the standard definition $\Delta_\omega^{(\mathrm{st})}$

$$\Delta_\omega = e^{\omega L} v^{-a\omega} \Delta_\omega^{(\mathrm{st})}. \tag{73}$$

For more details see Appendix B of [21] and also [26] as a review.

[5]The special alien derivatives $\Delta_\alpha$, $\Delta_\beta$ of $A_{\alpha,\beta}$ are in general non-vanishing (see Appendix C of [21]). By definition, they should appear in the Stokes automorphism as well, if the latter would act on $A_{\alpha,\beta}$. For $A_{\alpha;\ell}$ it also means that $\Delta_\alpha$ could be non-vanishing, and thus we had to include it in $\mathfrak{S}$. However for the $\alpha,\beta$-independent object $A_{0;\ell}$, they should have no effect at all.

[6]This is in contrast to the energy density of the relativistic $O(3)$ non-linear sigma model, which can be extracted from $\mathcal{O}_{1,1}$. In this case, the last 3 terms of (54) give exponentially suppressed real contributions that cannot be recovered from the perturbative part $A_{1,1}$ by asymptotic analysis.

integral equation (2) and the moments $\phi_\ell^{\mathrm{phys}}$ obtained from that solution. Our analysis strongly suggests complete agreement between the two:

$$S^+(\phi_\ell) = \phi_\ell^{\mathrm{phys}}, \tag{78}$$

up to the order $10^{-96}$ at $b = 10$.

Note that applying the lateral Borel resummation $S^+$ on the l.h.s. of (77) is then equivalent to the so-called median resummation of the asymptotic series $A_{0;\ell}$. That is, we find that the moments $\phi_\ell$ are resurgent in the strong sense [15, 17], and their perturbative data contains all the information needed to recover their physical value.

# 5 Capacitance of a circular parallel plate capacitor

The Lieb–Liniger integral equation is mathematically related to the famous Maxwell–Kirchhoff disk capacitor problem of electrostatics (see a historical review in [17]). In spite of the fact that it is more than one and a half centuries old, and seemingly theoretical in nature, the latter has relevance even in the present day applications, e.g. [27]. For this problem we can also present now a complete analytic trans-series solution.

We want to calculate the capacitance of two ideally thin conducting disks of radius $r$ in vacuum, arranged coaxially at a distance $d$ from each other. Love [28] reduced the problem to finding a solution to the integral equation (26), where the integration interval in (26) is related to the geometry of the problem by $b/\pi = r/d$. The solution to the integral equation encodes the surface charge density of the plates [29]. The capacitance $C$ (in SI units) is then directly related to the density $\mathcal{O}_{0,0}$:

$$C = 4\epsilon_0 d\, \mathcal{O}_{0,0}, \tag{79}$$

where $\epsilon_0$ is the vacuum permittivity.

From (55), we can build the trans-series of this density as [21, 22]

$$\mathcal{O}_{0,0} = \frac{1}{\pi}\hat{A}_{0,0} = \frac{1}{\pi}\Big[A_{0,0} + e^{-4b}S_2 A_{0,-2}^2 + e^{-8b}\big(S_2^2 A_{0,-2}^2 A_{-2,-2} + S_4 A_{0,-4}^2\big) + \mathcal{O}\big(e^{-12b}\big)\Big]. \tag{80}$$

Using the above expression, we can construct the trans-series of the capacitance in terms of the coupling $v$ defined in (24). However, we would like to write our results in terms of the variables $d$ and $r$ that define the geometry of the capacitor. Explicitly, we consider the ratio of the distance to the circumference of the disk capacitor:

$$\delta \equiv \frac{d}{2\pi r}. \tag{81}$$

The relation between the coupling $v$ and the ratio $\delta$ can be derived from (24):

$$\frac{1}{\delta} = 2b = \frac{1}{v} + \log\Big(\frac{v}{8e}\Big) \tag{82}$$

This expression can be inverted to yield

$$v = \frac{-1}{W_{-1}\big(-\frac{1}{8e}e^{-1/\delta}\big)} = \delta + O(\delta^2), \quad 0 \le v \lesssim 0.21706, \tag{83}$$

where $W_k(z)$ is the $k^{\mathrm{th}}$ branch of the Lambert $W$ function.

Using the relation (83), the trans-series in (80) for the capacitance can now be written in powers of the small parameters $\delta$ and $e^{-2/\delta}$, with the price that this expansion will also contain logarithms $\mathsf{L} \equiv -\log(\delta/8)$. We obtain the result

$$C = \epsilon_0 \frac{\pi r^2}{d}\Big[C^{(0)}(\delta, \mathsf{L}) + \delta \sum_{k=0}^{\infty} C^{(k)}(\delta, \mathsf{L}) e^{-2k(1/\delta+1)}\Big], \tag{84}$$

where the prefactor is the leading order result at small $\delta$, which corresponds to the well-known formula for the parallel-plate capacitor with small separation. The first few coefficients are given by

$$C^{(0)}(\delta, \mathsf{L}) = 1 + 2(\mathsf{L} - 1)\delta + (\mathsf{L}^2 - 2)\delta^2 + \frac{1}{2}(2\mathsf{L}^2 - 1 - 3\zeta_3)\delta^3 + O(\delta^4),$$

$$C^{(1)}(\delta, \mathsf{L}) = i\left\{2 + \left[2\mathsf{L} + \frac{3}{2}\right]\delta + \left[2\mathsf{L} + \frac{17}{16}\right]\delta^2 + \left[\frac{3}{2}\zeta_3 + \mathsf{L}\left(\frac{15}{16} - \mathsf{L}\right) + \frac{161}{192}\right]\delta^3 + O(\delta^4)\right\},$$

$$C^{(2)}(\delta, \mathsf{L}) = (1 + 4i) + \left((1 + 4i)\mathsf{L} + \left(\frac{1}{2} + \frac{3i}{2}\right)\right)\delta + \left[(1 + 4i)\mathsf{L} + \left(\frac{1}{4} + \frac{37i}{32}\right)\right]\delta^2$$

$$+ \left[\left(\frac{3}{4} + 3i\right)\zeta_3 - \mathsf{L}\left(\left(\frac{1}{2} + 2i\right)\mathsf{L} - \left(\frac{3}{4} + \frac{91i}{32}\right)\right) + \left(\frac{43}{96} + \frac{1301i}{768}\right)\right]\delta^3 + O(\delta^4), \quad (85)$$

$$C^{(3)}(\delta, \mathsf{L}) = \left(\frac{16}{3} + 13i\right) + \left[\left(\frac{16}{3} + 13i\right)\mathsf{L} + \left(2 + \frac{39}{12}i\right)\right]\delta$$

$$+ \left[\left(\frac{16}{3} + 13i\right)\mathsf{L} + \left(\frac{7}{8} + \frac{85i}{32}\right)\right]\delta^2$$

$$+ \left[\left(4 + \frac{39i}{4}\right)\zeta_3 - \mathsf{L}\left(\left(\frac{8}{3} + \frac{13i}{2}\right)\mathsf{L} - \left(\frac{107}{24} + \frac{331i}{32}\right)\right) + \left(\frac{383}{192} + \frac{637i}{128}\right)\right]\delta^3 + O(\delta^4).$$

Note that the perturbative part was already presented in a similar way in [12], including higher orders. The leading exponential correction was already obtained in [17], and the first few coefficients of the real part of the next-to-leading one were measured.[7]

## 6 The moments in the physical coupling $g$

Up to now, all results have been computed in terms of the coupling $v$ defined in (24). In this coupling, the exponential corrections to the moments can be computed with the expressions (63) and (55). However, we would like to rewrite the trans-series in terms of the physical coupling defined in (3).

By definition of the coupling, the zeroth moment can be written in terms of the physical coupling as

$$\phi_0 = \frac{1}{\pi g^2}. \tag{86}$$

Higher moments will be written in terms of $g$ by first deriving a trans-series relation between the original coupling $v$ and the physical coupling $g$:

$$v = \sum_{s=0}^{\infty} e^{-4n/g} \sum_{k=2-2n-\delta_{n,0}}^{\infty} v_k^{(n)} g^k. \tag{87}$$

The coefficients $v_k^{(n)}$ can be fixed by plugging the above ansatz in the trans-series of $\phi_0$, given in (71), and then imposing the constraint (86). In this way, we obtain a system of equations for the coefficients $v_k^{(n)}$. For example, imposing the condition (86) on the coefficients of $g^{-2}$, $g^{-1}$ and $g^0$, we find

$$\frac{1}{4(v_1^{(0)})^2} = 1, \quad \frac{2(v_1^{(0)})^2 + v_2^{(0)}}{2(v_1^{(0)})^3} = 0, \quad \frac{(v_1^{(0)})^4 - 4v_2^{(0)}(v_1^{(0)})^2 - 3(v_2^{(0)})^2 + 2v_3^{(0)}v_1^{(0)}}{4(v_1^{(0)})^4} = 0, \quad (88)$$

---

[7]Typos in exactly these formulas (4.29 and 4.30) are present in the published version of [17], however the analysis in the running coupling is correct.

which can be solved for $v_1^{(0)}$, $v_2^{(0)}$ and $v_3^{(0)}$. Extending the computation to further powers and exponential corrections in $g$, we find the trans-series

$$
\begin{aligned}
v = \frac{g}{2} - \frac{g^2}{2} &+ \frac{3g^3}{16} + \frac{9 - 3\zeta_3}{64}g^4 + \frac{-137 + 42\zeta_3}{768}g^5 + O(g^6) \\
&+ \left[\frac{64i}{e^4} + \frac{104i}{e^4}g + i\frac{-121 + 48\zeta_3}{2e^4}g^2 - i\frac{1577 + 912\zeta_3}{48e^4}g^3 + O(g^4)\right]e^{-4/g} \\
&\quad + \left[\frac{-24576 + 32768i}{e^8g^2} + \frac{34816 - 75776i}{e^8g}\right. \\
&\quad + \left.\frac{(-18944 + 90432i) + (18432 - 24576i)\zeta_3}{e^8} + O(g)\right]e^{-8/g} + O\left(e^{-12/g}\right). \quad (89)
\end{aligned}
$$

Finally, we can use this result to reexpress any trans-series from the $v$ coupling to the $g$ coupling. For example, the trans-series of $\phi_1$, given in (72), can be rewritten as

$$
\begin{aligned}
\phi_1 = \frac{1}{\pi^3}\left[\frac{1}{4g^4}\right. &- \frac{2}{3g^3} + \frac{\pi^2 - 6}{6g^2} + \frac{3\zeta_3 - 4}{4g} + \frac{3\zeta_3 - 4}{6} + O(g^1) \\
&+ \left(-\frac{128i}{e^4g^4} - \frac{80i}{e^4g^3} + i\frac{48\zeta_3 - 97}{e^4g^2} + i\frac{1104\zeta_3 - 1643}{24e^4g} + O(g^0)\right)e^{-4/g} \\
&+ \left(-\frac{32768 - 32768i}{e^8g^6} - \frac{4096 + 14336i}{e^8g^5} + \frac{(-24576 + 24576i)\zeta_3 + (22528 - 33088i)}{e^8g^4}\right. \\
&\quad + \left.\left.\frac{(33792 + 7680i)\zeta_3 + (-21632 - 9940i)}{3e^8g^3} + O(g^{-2})\right)e^{-8/g} + O\left(e^{-12/g}\right)\right]. \quad (90)
\end{aligned}
$$

We were able to compute the moments $\phi_1$, $\phi_2$, $\phi_3$, $\phi_4$ and $\phi_5$ with up to 20 exponential corrections, with 20 terms in each exponential correction. It is substantially easier to compute the coefficients numerically (even at high precision), in which case we obtained 20 exponential corrections, each with 336 terms.[8]

# 7 Numerical investigations

In this section we provide extensive checks on our trans-series solution for $\phi_\ell$. We test both the imaginary and real parts of it.

For the first we use the theory of resurgent functions and verify the coefficients of the leading, purely imaginary exponential contribution of $\phi_1$ from the large order asymptotics of the coefficients of the perturbative part $\phi_1^{(0)}$ in Subsection 7.1.

For the second we take the resummations of the perturbative part and the leading correction with the so-called Pade–Borel technique, and subtract it from a high-precision direct evaluation of the TBA in Subsubsection 7.2.1. We show that this difference is compatible with the real part of the coefficients of the next-to-leading exponential contribution. Then calculating similar subtractions order-by-order in exponential terms we show that the resummed trans-series indeed approximates the exact solution exponentially better at each step in Subsubsection 7.2.2, also in terms of the physical coupling $g$. Finally we repeat this latter analysis for some of the lowest higher moments $\phi_\ell$.

## 7.1 Asymptotic analysis

Here we would like to verify formula (75) for $\phi_1^{(0)} = 4\pi^{-3}A_{0;1}$, that is

$$
\frac{\pi^3}{4}\Delta_2\phi_1^{(0)} = 2S_2A_{0,-2}A_{-2;1} \quad (91)
$$

---

[8]For the numerical investigations in Section 7 the first 6 exponential order proved to be sufficient, as the precision was limited by the Borel resummations' error around this order of magnitude.

numerically, as a demonstration and a consistency check. The moments $A_{0,-2}$ and $A_{-2;1}$ can be obtained from the differential equations (31)-(34) perturbatively as

$$A_{0,-2} = -\frac{1}{4\sqrt{v}}\left\{1 - \frac{v}{8} - \frac{31}{128}v^2 + v^3\left(\frac{3}{8}\zeta_3 - \frac{937}{3072}\right) + v^4\left(\frac{11}{4}\zeta_3 - \frac{17397}{32768}\right)\right.$$
$$\left. + v^5\left(\frac{12957}{1024}\zeta_3 + \frac{405}{128}\zeta_5 - \frac{3613649}{3932160}\right) + O(v^6)\right\} \tag{92}$$

and

$$A_{-2;1} = -\frac{1}{32\sqrt{v}}\left\{\frac{1}{v^2} - \frac{49}{8v} + \left(32\varphi_{1,2}^{(0)} - \frac{2367}{128}\right)\right.$$
$$+ v\left(-8\varphi_{1,2}^{(0)} + \frac{39}{8}\zeta_3 + \frac{5447}{3072}\right) + v^2\left(-\frac{31}{2}\varphi_{1,2}^{(0)} + \frac{79}{16}\zeta_3 + \frac{585185}{98304}\right)$$
$$\left. + v^3\left(\left(24\zeta_3 - \frac{937}{48}\right)\varphi_{1,2}^{(0)} + \frac{1305}{128}\zeta_5 - \frac{9407}{1024}\zeta_3 + \frac{34943551}{3932160}\right) + O(v^4)\right\}, \tag{93}$$

where $\varphi_{1,2}^{(0)} = \frac{15}{32} + \frac{\zeta_2}{4}$ is the constant that is not determined by the differential equations (see the discussion at the end of Subsection (3.2))

Since $S_2 = 2ie^{-2}$, we obtain the r.h.s. of (91) as

$$2S_2 A_{0,-2} A_{-2;1} = \frac{i}{32e^2}\left\{\frac{1}{v^3} - \frac{25}{4v^2} + \frac{8\zeta_2 - \frac{95}{32}}{v}\right.$$
$$+ \frac{1}{384}(-768\zeta_2 + 2016\zeta_3 + 581) + \frac{v(-23040\zeta_2 + 29376\zeta_3 + 27745)}{6144} +$$
$$\left. v^2\left(6\zeta_2\left(\zeta_3 - \frac{211}{288}\right) + \frac{855\zeta_5}{64} - \frac{695\zeta_3}{64} + \frac{914731}{122880}\right) + O(v^3)\right\} \tag{94}$$

To measure the l.h.s. numerically, we calculated 100 coefficients of the perturbative series $\phi_1^{(0)}$, and analysed their asymptotics. Their structure was fitted as

$$\varphi_{1,n}^{(0)} \sim Y_0\frac{\Gamma_{n+5}}{2^{n+5}} + Y_1\frac{\Gamma_{n+4}}{2^{n+4}} + Y_2\frac{\Gamma_{n+3}}{2^{n+3}} + Y_3\frac{\Gamma_{n+2}}{2^{n+2}} + Y_4\frac{\Gamma_{n+1}}{2^{n+1}} + Y_5\frac{\Gamma_n}{2^n} + \dots \tag{95}$$

where $\Gamma_{n+j} \equiv \Gamma(n+j)$, and the correct integer shift $j$ in the factorial can be also measured.[9]

Using the definition of the Borel transform given Appendix B of [21]

$$\Psi(v) \sim \sum_{n \geq -N_{\min}} \psi_n v^n, \quad \Rightarrow \quad \mathcal{B}(t) = \sum_{n \geq 0} \frac{\psi_{n+1}}{\Gamma_{n+1}}t^n \tag{96}$$

we arrive at

$$\mathcal{B}(t) = Y_0\frac{5!}{(t-2)^6} - Y_1\frac{4!}{(t-2)^5} + Y_2\frac{3!}{(t-2)^4} - Y_3\frac{2!}{(t-2)^3} + Y_4\frac{1!}{(t-2)^2} - Y_5\frac{0!}{(t-2)^1} + \dots, \tag{97}$$

where each term in (95) - up to this order - corresponds to a (higher order) pole on the Borel plane. The alien derivative at the closest singularity to the origin is related to the difference of the lateral Borel resummations

$$S^{\pm}(\Psi)(v) = \sum_{n \geq -N_{\min}}^{0} \psi_n v^n + \int_{C_{\pm}} \mathrm{d}t\, e^{-t/v}\mathcal{B}(t), \tag{98}$$

---

[9]These terms corresponds to poles (and for $j < 0$ to logarithmic cuts) on the Borel-plane - see (97), and eventually give rise to powers of $v$ in the asymptotic expansion of the alien derivative (100).

that is, to

$$e^{-2/v}S^+(\Delta_2^{(\mathrm{st})}\phi_1^{(0)}) = S^+(\phi_1^{(0)}) - S^-(\phi_1^{(0)}) = \left(\int_{C_+} - \int_{C_-}\right)\mathrm{d}t\, e^{-t/v}\mathcal{B}(t), \qquad (99)$$

where the $C_\pm$ contours are running infinitesimally above and below the positive real axis. In this case the difference of the contours can be shrunk around the singularity at $t = 2$ and the residues give the (asymptotic) expansion of the alien derivative

$$\Delta_2^{(\mathrm{st})}\phi_1^{(0)} \sim -2\pi i\left(\sum_{k=0}^{5} Y_k v^{k-5} + O(v)\right). \qquad (100)$$

Converting the above, standard definition of the alien derivative at $\omega = 2$ to the one introduced in (75) (see the footnote there) we get

$$\Delta_2 = \left(\frac{v}{8e}\right)^2 \Delta_2^{(st)}. \qquad (101)$$

We have then

$$\Delta\phi_1^{(0)} \sim -\frac{i\pi Y_0}{2^5 e^2}\left(\sum_{k=0}^{5}(Y_k/Y_0)v^{k-3} + O(v^3)\right). \qquad (102)$$

To test whether $Y_0$ and the ratios $Y_k/Y_0$ agree with the coefficients that can be read off from (94), we used the method as sketched below.

We divided the coefficients with the leading asymptotics $\Gamma_{n+5}2^{-n}$ in (95), and used Richardson extrapolation [30] as a series acceleration method to extract the constant asymptotics. That is, if one has a series $x_n$ known up to a certain order, one can recursively compute $x_n^{(s)}$

$$x_n^{(s)} = \frac{1}{s}\left((n+1-s)x_n^{(s-1)} - (n+1-2s)x_{n-1}^{(s-1)}\right), \quad x_n^{(0)} \equiv x_n \qquad (103)$$

where with each step a correction term will drop out to the constant asymptotics in powers of $n^{-1}$:

$$x_n = \mathrm{const.} + O(n^{-1}) \quad \Rightarrow \quad x_n^{(s)} = \mathrm{const.} + O(n^{-s-1}). \qquad (104)$$

Then the last available term of the acceleration $x_n^{(s)}$ is a good estimate of the constant. The order $s$ can be optimized, however for this analysis we used $s = 30$ in every case.

After measuring $Y_0$ in this way, we can confirm that it indeed agrees with the exact value $Y_0 = -4\pi^{-4}$ that can be deduced by comparing (94) and (102) - see Table 1. Knowing this exact value of $Y_0$ the leading term can be subtracted, and division by $Y_0\Gamma_{n+4}2^{-n-4}$ revealed the ratio $Y_1/Y_0$ after we used Richardson extrapolation again. This ratio again agrees with the exact value $Y_1/Y_0 = -25/4$ up to high precision. These subtractions can then be repeated for the subleading terms and this allows us to measure $Y_k/Y_0$ up to a certain order, that is limited by the number of coefficients known in $\varphi_{1,n}^{(0)}$ and their precision. For the subtractions at each step we were using the known exact values from (94) instead of the measured ones, to achieve higher precision. Nevertheless, the fact that the measured and exact values match, up to several digits, confirms our analytical result (75).

## 7.2 Comparison to TBA

### 7.2.1 In the bootstrap coupling $v$

At first we provide our readers with a fast and technically simpler check on whether our transseries indeed matches the solution of the integral equation. This analysis goes only up to the

| | Measured value | Exact value |
|---|---|---|
| $Y_0$ | $-0.04106392901873{\scriptstyle 8}$ | $-4\pi^{-4}$ |
| $Y_1/Y_0$ | $-6.249999999999{\scriptstyle 3}$ | $-25/4$ |
| $Y_2/Y_0$ | $10.19072253478{\scriptstyle 8}$ | $4\pi^2/3 - 95/32$ |
| $Y_3/Y_0$ | $4.533951440$ | $21\zeta_3/4 - \pi^2/3 + 581/384$ |
| $Y_4/Y_0$ | $4.09461953$ | $153\zeta_3/32 - 5\pi^2/8 + 27745/6144$ |
| $Y_5/Y_0$ | $12.87618$ | $855\zeta_5/64 + \pi^2(\zeta_3 - 211/288) - 695\zeta_3/64 + 914731/122880$ |

Table 1: The asymptotic coefficients of $\phi_1^{(0)}$ as measured by the last value in the $30^{\text{th}}$ Richardson accelerant, shown up to the digit they match the exact values of the coefficients in $\Delta_2\phi_1^{(0)}$. The first differing digit is also shown in smaller font.

second non-perturbative correction of $\phi_1$

$$\phi_1 = \frac{4}{\pi^3}\left\{ A_{0;1} + S_2 A_{0,-2} A_{-2;1} e^{-4b} \right.$$

$$\left. + \left(S_2^2 A_{0,-2} A_{-2,-2} A_{-2;1} + S_4 A_{0,-4} A_{-4;1}\right) e^{-8b} + O\left(e^{-12b}\right) \right\}. \quad (105)$$

We calculated the perturbative series of $\phi_1$ and its first non-perturbative correction—the second term on the r.h.s. of (105)—up to high orders numerically. That is, using the differential relations (31)–(34) from the previously generated 336 coefficients [16] obtained for the energy density (i.e. $A_{1,1}$) of the $O(3)$ non-linear sigma model via Volin's method. This requires using the relations on a route that is different from that sketched in Subsection 3.1 and looks as:

$$A_{1,1} \overset{(31)}{\to} a_1 \overset{(34)}{\to} a_\alpha \to \begin{cases} A_{\alpha,\beta} & (31) \text{ or } (32), \\ A_{\alpha,0} & (31) \text{ or } (32), \\ A_{\alpha;\ell} & (35) \text{ and } (36). \end{cases} \quad (106)$$

We then resummed both the perturbative part and the first non-perturbative correction via the Borel–Padé method. That is, after Borel transforming the asymptotic series with finitely many numerical coefficient, we took its diagonal Padé-approximant, which captures the analytic structure of the function on the Borel plane. Then, we applied lateral Borel resummation by numerically integrating the approximant along a semi-infinite line at a finite acute angle to the positive real line. See e.g. Section 12 of [21] for more details.

The real part of this resummation compares to the high-precision numerical solution of the TBA. This was obtained with the very efficient method developed in [31]. It is based on expanding the unknown function of the integral equation on a truncated basis of even Chebyshev polinomials, and solving a linear system for its expansion coefficients. TBA equations of the same type can be solved by similar methods for other models [30] too. The efficiency of the technique in [31] lies in a recursion relation that can be used to accelerate the calculation of the Lieb–Liniger kernel on the Chebyshev basis. With this method we were able to obtain numeric solutions in the range $1 \le b \le 13$ with at least 300 digits of precision even for the upper end of this range, where the algorithm produces the largest relative errors due to the truncation of the basis.

The difference of the lateral Borel resummation and the numerical TBA is shown in Figure 1. The data points correspond to the values $b = 1, 2, \ldots, 13$. According to (105), this difference can be approximated by the real part of the lateral resummation of the third term on the r.h.s., which corresponds to the contribution proportional to $S_2^2$,[10] and has the following asymptotic

---

[10]The contribution proportional to $S_4$ in (105) is purely imaginary.

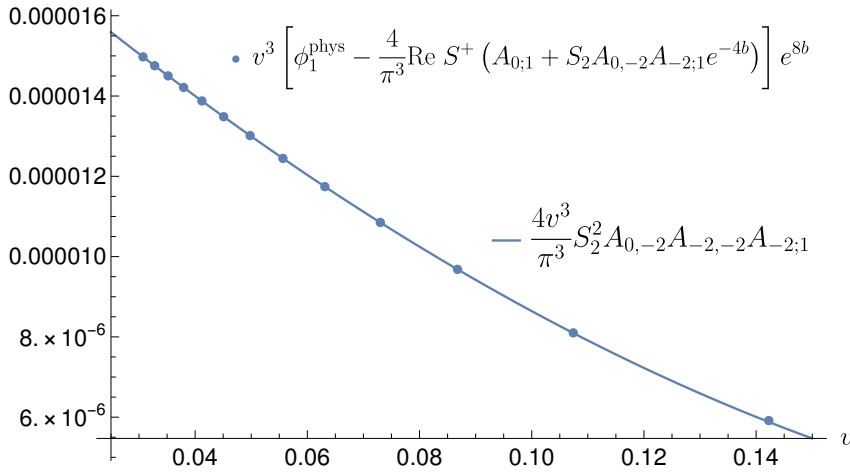

Figure 1: The plot shows that the difference of the resummed trans-series up to the first non-perturbative order matches the real part contribution of the second non-perturbative order. We plotted the terms shown in (107) as the solid curve. Note that on the range of the plot, only the few leading terms of (107) can be verified precisely, approximately up to the $O(1)$ contribution.

expansion:

$$S_2^2 A_{0,-2} A_{-2,-2} A_{-2;1} = \frac{1}{128e^4} \left\{ \frac{1}{v^3} - \frac{13}{2v^2} + \frac{8\zeta_2 - \frac{7}{4}}{v} \right. \tag{107}$$
$$+ \left( -4\zeta_2 + \frac{21\zeta_3}{4} + \frac{379}{96} \right) + \left( -6\zeta_2 + \frac{69\zeta_3}{16} + \frac{2797}{384} \right) v$$
$$\left. + \left( \zeta_2 \left( 6\zeta_3 - \frac{77}{12} \right) + \frac{-20820\zeta_3 + 25650\zeta_5 + 19403}{1920} \right) v^2 + O(v^3) \right\}.$$

In the plot, the difference between the lateral Borel resumation and the numerical TBA is thus divided by its expected magnitude $e^{-8b}$, and the second non-perturbative correction (the truncated series in (107)) is plotted against this difference, shown as the solid line. In the range shown, the two results agree.

To obtain the expansion in (107), we needed to calculate yet another building block in addition to the ones shown in Subsection 7.1, namely

$$A_{-2,-2} = -\frac{1}{4} + \frac{v}{16} + \frac{11v^2}{128} + \frac{175v^3}{1536} + \left( \frac{4439}{24576} - \frac{27}{128}\zeta_3 \right) v^4 \tag{108}$$
$$+ \left( \frac{191429}{491520} - \frac{1061}{512}\zeta_3 \right) v^5 + \left( -\frac{45119}{4096}\zeta_3 - \frac{3375}{1024}\zeta_5 + \frac{8556971}{11796480} \right) v^6 + O(v^7).$$

### 7.2.2 In the physical coupling $g$

As mentioned in Section 6, we also calculated the trans-series (63) in the coupling $g$ for the observables $\phi_\ell$, up to several exponential corrections with 336 perturbative coefficient in each. This was done numerically, with many ($\sim 2500$) digits of precision using the same dataset that we used in Subsubsection 7.2.1.

For demonstrating how our trans-series solution can approximate the physical value, we chose the normalized quantity

$$\epsilon(g) \equiv 4\pi^3 g^4 \phi_1 = 1 - \frac{8}{3}g + O(g^2), \tag{109}$$

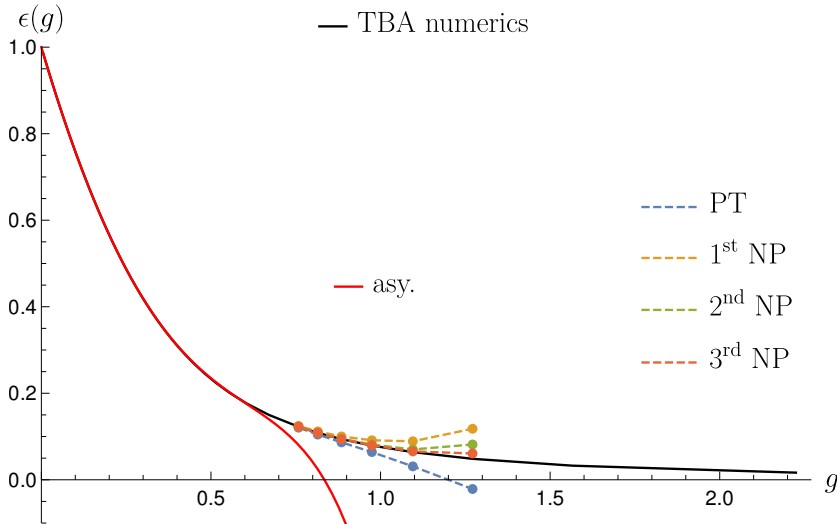

Figure 2: The resummations including more and more non-perturbative ("NP") corrections (dashed lines) approximate the high precision numerical value of the TBA (solid black line) increasingly better, than using only the perturbative part ("PT"). The solid red line shows the asymptotic series ("asy.") of the perturbative part $\epsilon^{(0)}(g)$ truncated at the $g^7$ term. The Borel integrals were performed at the specific $g$ values corresponding to $b = 0.15, 0.2, 0.25, 0.3, 0.35, 0.4$ (see the dots on the dashed lines).

which was also defined in eq. (12) of [31]. We define the transmonomials $\epsilon^{(k)}(g)$ as

$$\epsilon(g) = \epsilon^{(0)}(g) + \epsilon^{(1)}(g)e^{-4/g} + \epsilon^{(2)}(g)e^{-8/g} + \epsilon^{(3)}(g)e^{-12/g} + O\left(e^{-16/g}\right), \qquad (110)$$

for which we can compute the asymptotic series of each. With the Borel–Padé method, we resummed the explicitly shown non-perturbative orders in (110) at a few $g$ values and plotted the result against a high-precision TBA numerical computation. The latter was obtained for several $b$ values in the range $0.05 \leq b \leq 200$ with the method mentioned in Subsubsection 7.2.1, to get a clear picture on the behaviour of the normalized moment $\epsilon(g)$. In Figure 2, as we include more non-perturbative corrections, the trans-series approximation gets closer and closer to the physical result.

To further demonstrate our analysis for higher moments $\phi_1, \phi_2, \ldots, \phi_5$, and validate our trans-series solution with greater confidence, we compare the resummations of several exponential corrections to the TBA value of these moments at a single value of the coupling. We use the value $g \simeq 0.082485$, which corresponds to $b = 10$. In this case $e^{-4/g} \simeq 8.7 \times 10^{-22}$ and thus we expect the exponential corrections to give only slight improvements on the value of the quantities.[11] Typically, for each exponential order that we include, the difference to the physical result is of the order of the next exponential correction, except for the $O(e^{-4/g})$ term, whose coefficients in the asymptotic expansion are purely imaginary, and thus only give real contributions after resummation at higher exponential orders.

At the given value $b = 10$, the estimated relative error for the moments using the numerical TBA result[12] was of the order of $10^{-352}$. According to numerical studies in [21], the error of the trans-series resummation is dominated by the lateral Borel integration of the perturbative part, and we estimated it (simply comparing resummations from 336 and 334 coefficients) to be of the order of $10^{-101}$.

---

[11]The reason behind that the magnitude of the consecutive exponential corrections in Table 2 are not really on the order of the powers of this numerical value of $e^{-4/g}$ is the fact that each exponential correction in the trans-series typically starts with a high inverse power of $g$, and thus the values of these asymptotic series (even after resummation) is quite large at this small $g$ coupling.

[12]The Chebyshev basis of method [31] was truncated at the 1400th polynomial.

| $N$ | $\epsilon_{\text{TBA}} - \text{Re} \sum_{n=0}^{N} S^+(\epsilon^{(n)})e^{-4n/g}$ | $\text{Re } S^+(\epsilon^{(N+1)})e^{-4(N+1)/g}$ | |
|---|---|---|---|
| 0 | $4.933013647272668\underline{9303} \times 10^{-39}$ | $2 \times \quad 4.933013647272668\underline{9247} \times 10^{-39}$ | ✓ |
| 1 | $-4.933013647272668\underline{9192} \times 10^{-39}$ | $- \quad 4.933013647272668\underline{9082} \times 10^{-39}$ | ✓ |
| 2 | $-1.10595929473457\underline{8167} \times 10^{-56}$ | $- \quad 1.10595929473457\underline{8146} \times 10^{-56}$ | ✓ |
| 3 | $-2.09363599462608\underline{24762} \times 10^{-74}$ | $- \quad 2.09363599462608\underline{24735} \times 10^{-74}$ | ✓ |
| 4 | $-2.7563259310\underline{022551778} \times 10^{-92}$ | $- \quad 2.7563259317\underline{706920524} \times 10^{-92}$ | ✓ |
| 5 | $\underline{7.684368746700002357}6 \times \boxed{10^{-102}}$ | $\underline{1.892493353367489874}7 \times \boxed{10^{-110}}$ | ⚡ |

Table 2: Left column: the difference between the resummed trans-series of $\epsilon(g)$ truncated at the $N^{\text{th}}$ exponential correction and the high-precision numerics $\epsilon_{\text{TBA}}(g)$. Right column: the next (missing) exponential correction is shown for comparison. They typically match up to several orders of magnitude (differing digits are indicated by underlining), except for the $N = 0$ and $N = 5$ rows for different reasons (see the main text).

In Table 2, we show the difference between the physical $\epsilon$ value

$$\epsilon(g)\big|_{b=10} = 0.79735795625339861330970010774448648181804542381949075975865936 2497$$
$$709063650575812428669484098728005702\ldots \tag{111}$$

and the resummed trans-series truncated at different exponential orders. These results are then compared to the next exponential correction in the trans-series. The conclusion we can draw from it is that the next exponential order accounts for the missing part in the trans-series, at least up to the order of the following exponential correction.[13] With this we could show that at least up to $O(e^{-20/g})$ our trans-series for $\phi_1$ must be correct. After subtracting also the $5^{\text{th}}$ exponential correction, we bump into the error level of the Borel–Padé technique for the perturbative part at the known number of perturbative coefficients, as discussed above. This is clear from the last line of Table 2, as the magnitude of the $6^{\text{th}}$ exponential correction is much smaller.

For the higher moments $\phi_\ell$ we define similar normalizations $\epsilon_\ell$ as we did for $\phi_1$ (such that their small $g$ expansion would start with 1):

$$\phi_\ell \equiv \frac{2\pi(-1)^\ell}{(\pi g)^{2(\ell+1)}} \binom{1/2}{\ell+1} \epsilon_\ell. \tag{112}$$

We performed the numerical analysis for the other moments as in the case of $\epsilon(g) \equiv \epsilon_1(g)$. However, for simplicity we only present the subtraction of the first $N = 4$ exponential terms of the trans-series for these quantities in Table 3. The values show agreement at least up to the order $10^{-96}$.

---

[13]This picture gets further complicated by the fact that the coefficients of the $e^{-4/g}$ term in the trans-series are purely imaginary, and thus its resummation contributes to the real part only at $O(e^{-8/g})$. This contribution is twice the magnitude of the difference of the TBA and the resummation of the perturbative part (as indicated by the factor $2\times$ in the $N = 0$ row of Table 2). Thus including it in the sum over non-perturbative terms flips the sign of the difference, and then only the second, $O(e^{-8/g})$ term of the tran-series (whose coefficients have already real parts) can drop this contribution out, reducing the difference to $O(e^{-12/g})$.

| $\ell$ | $\epsilon_\ell^{\text{TBA}} - \text{Re } \sum_{n=0}^{4} S^+(\epsilon_\ell^{(n)}) e^{-4n/g}$ | $\text{Re } S^+(\epsilon_\ell^{(5)}) e^{-20/g}$ |
|---|---|---|
| 1 | $-2.756325931\underline{0} \times 10^{-92}$ | $-2.756325931\underline{8} \times 10^{-92}$ |
| 2 | $-6.693787888\underline{0} \times 10^{-92}$ | $-6.693787888\underline{8} \times 10^{-92}$ |
| 3 | $-1.035566200\underline{3} \times 10^{-91}$ | $-1.035566199\underline{8} \times 10^{-91}$ |
| 4 | $-1.2942070\underline{432} \times 10^{-91}$ | $-1.294207\underline{0554} \times 10^{-91}$ |
| 5 | $-1.41698\underline{35167} \times 10^{-91}$ | $-1.4169\underline{863583} \times 10^{-91}$ |

Table 3: Difference between the TBA numerics and the resummation of the trans-series up to the $N = 4$ exponential order, for the higher (normalized) moments $\epsilon_\ell$. The values are presented along the $N = 5$ exponential order, in the manner of Table 2.

# 8    Conclusions

In this paper we presented a solution for the moments of the Lieb–Liniger model including perturbative and non-perturbative corrections, *i.e.* a trans-series solution. The perturbative parts of the moments [11, 12] can be extracted through a method devised by Volin in [13, 14]. Since the integral equation describing the rapidity density has support on the interval $[-B, B]$, the moments are expressed as a perturbative expansion in $1/B$ and also contain $\log(B)$ terms. It proved advantageous [17] to introduce a running coupling $v$, which leads to a cancellation of the $\log B$ terms.

The conserved charges of the Lieb–Liniger model are given by its moments. A second order differential equation [7] relates the moments $\phi_\ell$ and $\phi_{\ell-1}$, allowing to recursively generate the higher moments up to integration constants. In fact, there are two integration constants for each moment; one of these can be fixed through a constraint [7]. The remaining constant is the coefficient at order $1/v^2$ in the perturbative expansion and can be obtained by using Volin's method [13, 14]. With the constants fixed, we can compute $\phi_\ell$ up to arbitrary perturbative order.

Further, we used the methods developed in [18, 21, 22] for relativistic models to express the observables $\mathcal{O}_{\alpha,\beta}$ as a trans-series using a perturbatively defined basis $A_{\alpha,\beta}$. The perturbative basis $A_{\alpha,\beta}$ in turn satisfies a system of differential equations, that allow to construct all basis elements from one explicitly given element. This element, for instance $A_{0,0}$, can be determined using the algorithm of Volin. To write the trans-series solution for the moments, we need to introduce the generalized moments. This generalization completes the perturbative basis necessary to construct non-perturbative corrections to the moments. Moreover, we checked that the trans-series solution is indeed a solution of the set of ODEs for the observables (31)–(34) as well as (35) for the moments. Let us emphasize again, that this allows us to calculate any moment as a trans-series to arbitrary order in the coupling $v$ including its non-perturbative corrections in $e^{-1/v}$. As an application of the trans-series solution we discussed the disk capacitor problem, which can be related to the Lieb–Liniger integral equation.

The results we presented were mainly expressed in terms of the coupling $v$, as it allows to write equations in a more compact manner. However, the Lieb–Liniger model comes with a naturally defined physical coupling $g^2$ given in (3), that is proportional to the first moment. Expressing $v$ in terms of $g$ allowed us to reexpress any moment in terms of the physical coupling.

Finally, we turned to numerical investigations. By analyzing the asymptotic behavior of the generalized moment $\phi_1^{(0)}$, we could measure its alien derivative numerically, providing a consistency check of (75). Furthermore, we checked that the trans-series indeed matches the solution obtained from the thermodynamic Bethe ansatz with high precision, both in the couplings $v$ and $g$, respectively. Remarkably, including five non-perturbative orders, we found agreement at least up to the order $10^{-96}$.

In the recent work [8] the authors analyzed the rapidity distribution in the Lieb–Liniger model and derived exact relations for its derivatives at the Fermi level, which they evaluated at weak and strong couplings. Their weak coupling expansion in $g$ was an asymptotic series, which did not include exponentially suppressed corrections. Thus plugging our trans-series expressions into their formulas can provide the full non-perturbative results for the sound velocity, the Luttinger parameter or other interesting observables.

We would like to compare our results also to the interesting conjectures in [32, 33]. Based on partial resummations of patterns in the large $\gamma$ expansions of the energy density, the authors of these papers proposed a heuristic structure for the small $\gamma$ regime that is analytically different from the asymptotic expansions studied by us. The relation of these two types of representations was already pursued in [33] for the Gaudin–Yang model. Figuring out whether this connection can be extended to the trans-series level might improve on the understanding, how the weak and strong coupling regimes can be related analytically.

In relativistic models, the source of the analogous non-perturbative corrections is thought to have physical origin, either in terms of instantons [16, 19] or renormalons [34, 15]. The first has a semiclassical interpretation related to saddle points of the path integral, and there are attempts to relate renormalons to them as well [35, 36]. The trans-series of correlation functions was also connected to vacuum condensates [37]. It is then compelling to ask whether the corrections we calculated here can be attributed to semiclassical effects. For the case of the Gaudin–Yang model, the superconducting gap was identified as the source of the large order behaviour of the perturbative series, instead of the semiclassical features of the path integral [11]. Similar attempts were made out for the Lieb-Liniger model in [38]. Based on the investigation of its relativistic counterpart, the $O(N)$ symmetric $\phi^4$ theory, the authors argued that renormalons in two dimensional theories are related to the absence of Goldstone bosons (Coleman-Mermin-Wagner theorem). Performing perturbation theory around the wrong vacuum, IR divergences appear, which can be regulated and eventually cancelled for physical observables such as for the ground-state energy, [39], but whose imprint as renormalons is preserved. This was demonstrated at large $N$ by identifying a set of diagrams, which are IR finite, but contribute to the factorial growth of the perturbative expansion, see also [40]. In the Lieb-Liniger model the IR finiteness was also checked up to two loops and the Bethe ansatz result was recovered. However, it is not possible to perform a large $N$ calculation in the Lieb-Liniger model, and it would be challenging to identify diagrams, which could provide factorially growing perturbative coefficients. There are non-perturbative objects that arise as coherent structures in Bose-Einstein condensates, the so-called dark/bright solitons [41]. They were studied extensively in the Lieb–Liniger case [42, 43], as the classical solutions of the Gross-Pitaevskii/nonlinear Schrödinger equation that appears in the second quantized formalism of the bosons. It would be interesting to examine whether these solutions can have any relevance in the non-perturbative effects we obtained from the TBA description in this work.

## Acknowledgements

We thank Zoran Ristivojevic for the fruitful discussions at an early phase of this project and also for drawing our attention to the numerical method in [31] and sharing us its implementation. This work was supported by the NKFIH research grant K134946. DlP is supported through a Feodor Lynen Fellowship by the Alexander von Humboldt Foundation. The work of IV was also partially supported by the NKFIH excellence grant TKP2021-NKTA-64.

## A    More on the checks of the trans-series solution

In this appendix we will introduce some useful notation and fill in some details we omitted in Section 4.2. Moreover, we will explicitly show that the trans-series ansatz solves the differential

equations (33)–(35). The remaining equations (31) and (32) are not independent.

## A.1 Notation and identities

**Non-perturbative quantities.** To simplify the proofs, it is convenient to define the non-perturbative objects

$$\hat{C}_{\alpha,\beta} = e^{(\alpha+\beta)b}\hat{A}_{\alpha,\beta}, \qquad \hat{C}_{\alpha} = e^{\alpha b}\hat{a}_{\alpha}. \tag{113}$$

The densities and boundary values can be expressed as

$$\mathcal{O}_{\alpha,\beta} = \frac{1}{4\pi}\left[G_+(i\alpha)G_+(i\beta)\hat{C}_{\alpha,\beta} + G_-(i\alpha)G_+(i\beta)\hat{C}_{-\alpha,\beta}\right. \tag{114}$$

$$\left. + G_+(i\alpha)G_-(i\beta)\hat{C}_{\alpha,-\beta} + G_-(i\alpha)G_-(i\beta)\hat{C}_{-\alpha,-\beta}\right],$$

$$\chi_\alpha = \frac{1}{2}G_+(i\alpha)\hat{C}_\alpha + \frac{1}{2}G_-(i\alpha)\hat{C}_{-\alpha}. \tag{115}$$

Further, we introduce the shorthand notation

$$\bar{\lambda}_\kappa = S_\kappa\, e^{-2\kappa b} \quad ; \quad \lambda_\alpha = \sigma(i\alpha+0)\, e^{-2\alpha b} \tag{116}$$

with which we can rewrite (55) as

$$(\mathcal{A})_{-\kappa_r,-\kappa_s} = \sum_{N=-1}^{\infty} \sum_{j_1,j_2,\dots,j_N=1}^{\infty} \bar{\lambda}_{\kappa_r} A_{-\kappa_r,-\kappa_{j_1}} \bar{\lambda}_{\kappa_{j_1}} A_{-\kappa_{j_1},-\kappa_{j_2}} \dots \bar{\lambda}_{\kappa_{j_N}} A_{-\kappa_{j_N},-\kappa_s} \bar{\lambda}_{\kappa_s}. \tag{117}$$

Here the $N = -1$ term of $(\mathcal{A})_{-\kappa_r,-\kappa_s}$ is formally to be understood as $\bar{\lambda}_\kappa \delta_{-\kappa_r,-\kappa_s}$. Explicitly, $(\mathcal{A})_{-\kappa_r,-\kappa_s}$ reads

$$(\mathcal{A})_{-\kappa_r,-\kappa_s} = \bar{\lambda}_{\kappa_r}\delta_{-\kappa_r,-\kappa_s}$$

$$+ \sum_{N=0}^{\infty} \sum_{j_1,j_2,\dots,j_N=1}^{\infty} \bar{\lambda}_\kappa A_{-\kappa_r,-\kappa_{j_1}} \bar{\lambda}_{\kappa_{j_1}} A_{-\kappa_{j_1},-\kappa_{j_2}} \dots \bar{\lambda}_{\kappa_{j_N}} A_{-\kappa_{j_N},-\kappa_s} \bar{\lambda}_{\kappa_s}. \tag{118}$$

The trans-series for $\hat{A}_{\alpha,\beta}$ and $\hat{a}_\alpha$ can then be written as in (55) and (58), respectively.

**Perturbative $C$'s.** Equivalently, we define $C_{\alpha,\beta}$ and $C_\alpha$ as the perturbative parts of the above objects $\hat{C}$, respectively as

$$C_{\alpha,\beta} = e^{(\alpha+\beta)b}A_{\alpha,\beta}, \qquad C_\alpha = e^{\alpha b}a_\alpha. \tag{119}$$

We will work under the assumption that $C_{\alpha,\beta}$ and $C_\alpha$ satisfy the integral equations

$$\dot{C}_{\alpha,\beta} = C_\alpha C_\beta, \tag{120}$$

$$\ddot{C}_\alpha - \alpha^2 C_\alpha = f\, C_\alpha. \tag{121}$$

which is a direct consequence of (52).

Together with (55) these perturbative objects $C$ allow for a more compact notation as we can write

$$e^{(\alpha+\beta)b}\sum_{r,s} A_{\alpha,-\kappa_r}(\mathcal{A})_{-\kappa_r,-\kappa_s}A_{\kappa_s,\beta} =$$

$$\sum_{r,s} C_{\alpha,-\kappa_r}\underbrace{\left(\sum_{N=-1}^{\infty}\sum_{j_1,\dots,j_N=1}^{\infty} S_{\kappa_r}C_{-\kappa_r,-\kappa_{j_1}}S_{\kappa_{j_1}}C_{-\kappa_{j_1},-\kappa_{j_2}}\dots C_{-\kappa_{j_N},-\kappa_s}S_{\kappa_s}\right)}_{(\mathcal{C})_{-\kappa_r,-\kappa_s}}C_{-\kappa_s,\beta}. \tag{122}$$

which in turn is consistent with the definition given in (113) for $\hat{C}$.

Further, note that $\hat{C}_{\alpha,\beta}$ can be written as

$$
\begin{aligned}
\hat{C}_{\alpha,\beta} &= C_{\alpha,\beta} + \sum_{r,s} C_{\alpha,-\kappa_r}(\mathcal{C})_{-\kappa_r,-\kappa_s} C_{-\kappa_s,\beta} \\
&= C_{\alpha,\beta} + \sum_{s} \hat{C}_{\alpha,-\kappa_s} S_{\kappa_s} C_{-\kappa_s,\beta} \, ,
\end{aligned}
\tag{123}
$$

while for $\hat{C}_\alpha = e^{\alpha b}\hat{a}_\alpha$ we have

$$
\begin{aligned}
\hat{C}_\alpha &= C_\alpha + \sum_{r,s} C_{\alpha,-\kappa_r}(\mathcal{C})_{-\kappa_r,-\kappa_s} C_{-\kappa_s} \\
&= C_\alpha + \sum_{s} \hat{C}_{\alpha,-\kappa_s} S_{\kappa_s} C_{-\kappa_s} \, .
\end{aligned}
\tag{124}
$$

Finally, by definition $\hat{C}_{0,0} = \hat{A}_{0,0}$ and $\hat{C}_0 = \hat{a}_0$.

**Useful identities.** For $(\dot{\mathcal{C}})_{-\kappa,-\kappa'}$ we obtain

$$
\begin{aligned}
(\dot{\mathcal{C}})_{-\kappa,-\kappa'} &= \sum_{N=0}^{\infty} \sum_{j_1,\dots,j_N=1}^{\infty} \frac{\mathrm{d}}{\mathrm{d}b} S_\kappa C_{-\kappa,-\kappa_{j_1}} S_{\kappa_{j_1}} C_{-\kappa_{j_1},-\kappa_{j_2}} \dots C_{-\kappa_{j_N},-\kappa'} S_{\kappa'} \\
&= \sum_{N=0}^{\infty} \sum_{k=0}^{N} \sum_{j_1,\dots,j_N=1}^{\infty} S_\kappa C_{-\kappa,-\kappa_{j_1}} \dots \dot{C}_{-\kappa_{j_k},-\kappa_{j_{k+1}}} \dots C_{-\kappa_{j_N},-\kappa'} S_{\kappa'} \\
&= \sum_{N=0}^{\infty} \sum_{k=0}^{N} \left( \sum_{j_1,\dots,j_k=1}^{\infty} S_\kappa C_{-\kappa,-\kappa_{j_1}} \dots C_{-\kappa_{j_k}} \right) \left( \sum_{j_{k+1},\dots,j_N=1}^{\infty} C_{-\kappa_{j_{k+1}}} \dots C_{-\kappa_{j_N},-\kappa'} S_{\kappa'} \right) \\
&= \left[ \sum_{\hat{j}=1}^{\infty} (\mathcal{C})_{-\kappa,-\kappa_{\hat{j}}} C_{-\kappa_{\hat{j}}} \right] \left[ \sum_{\hat{j}=1}^{\infty} C_{-\kappa_{\hat{j}}} (\mathcal{C})_{-\kappa_{\hat{j}},-\kappa'} \right] = \hat{C}_{-\kappa} \hat{C}_{-\kappa'}
\end{aligned}
\tag{125}
$$

Since the building blocks $A_{\alpha,\beta}$ are symmetric in their indices, this is also true for $C_{\alpha,\beta}$ and hence the matrix $\mathcal{C}$. Therefore,

$$
\sum_{\hat{j}=1}^{\infty} C_{-\kappa_{\hat{j}}} (\mathcal{C})_{-\kappa_{\hat{j}},-\kappa'} = \sum_{\hat{j}=1}^{\infty} (\mathcal{C})_{-\kappa',-\kappa_{\hat{j}}} C_{-\kappa_{\hat{j}}} \, .
\tag{126}
$$

Further, we observe that

$$
\begin{aligned}
\frac{\mathrm{d}}{\mathrm{d}b} \hat{C}_{n,-\kappa_s} S_{\kappa_s} &= \frac{\mathrm{d}}{\mathrm{d}b} \sum_r (C_{n,-\kappa_r}(\mathcal{C})_{-\kappa_r,-\kappa_s}) \\
&= \sum_r C_{-n} C_{-\kappa_r}(\mathcal{C})_{-\kappa_r,-\kappa_s} + \sum_{r,t,u}(C_{n,-\kappa_r}(\mathcal{C})_{-\kappa_r,-\kappa_t} C_{-\kappa_t})(C_{-\kappa_u}(\mathcal{C})_{-\kappa_u,-\kappa_s}) \\
&= \hat{C}_n \hat{C}_{-\kappa_s} S_{\kappa_s} \, .
\end{aligned}
\tag{127}
$$

## A.2 The first equation

We start from the equation given in (32). Again we can plug in the trans-series solution (54) and since this equation holds for every coefficient $\sigma_\alpha^+$ it is sufficient to consider

$$
(\alpha^2 - \beta^2)\hat{C}_{\alpha,\beta} = \dot{\hat{C}}_\alpha \hat{C}_\beta - \dot{\hat{C}}_\beta \hat{C}_\alpha \, .
\tag{128}
$$

Again, we work under the assumption, that the perturbative part of this equation holds, which is given by

$$(\alpha^2 - \beta^2)C_{\alpha,\beta} = \dot{C}_\alpha C_\beta - \dot{C}_\beta C_\alpha \,. \tag{129}$$

We will now show that (128) holds by using (129) repeatedly. Let us begin by plugging (123) into (128) to rewrite it in terms of the perturbative $C$'s as[14]

$$(\alpha^2 - \beta^2)\Big[C_{\alpha,\beta} + \hat{C}_{\alpha,-\kappa_s}C_{-\kappa_s,\beta}\Big]$$
$$= \Big(\dot{C}_\alpha + \hat{C}_\alpha\hat{C}_{-\kappa_r}C_{-\kappa_r} + \hat{C}_{\alpha,-\kappa_r}\dot{C}_{-\kappa_r}\Big)\hat{C}_\beta - \Big(\dot{C}_\beta + \hat{C}_\beta\hat{C}_{-\kappa_r}C_{-\kappa_r} + \hat{C}_{\beta,-\kappa_r}\dot{C}_{-\kappa_r}\Big)\hat{C}_\alpha \tag{130}$$
$$= \Big(\dot{C}_\alpha + \hat{C}_{\alpha,-\kappa_r}\dot{C}_{-\kappa_r}\Big)\hat{C}_\beta - \Big(\dot{C}_\beta + \hat{C}_{\beta,-\kappa_r}\dot{C}_{-\kappa_r}\Big)\hat{C}_\alpha \,.$$

We can now use the perturbative part of the second equation (129) to drop $C_{\alpha,\beta}$ and obtain

$$(\alpha^2 - \beta^2)\hat{C}_{\alpha,-\kappa_s}C_{-\kappa_s,\beta} = \Big(\dot{C}_\alpha\hat{C}_{\beta,-\kappa_t}C_{-\kappa_t} + \hat{C}_{\alpha,-\kappa_r}\dot{C}_{-\kappa_r}(C_\beta + \hat{C}_{\beta,-\kappa_t}C_{-\kappa_t})\Big)$$
$$- \Big(\dot{C}_\beta\hat{C}_{\alpha,-\kappa_t}C_{-\kappa_t} + \hat{C}_{\beta,-\kappa_r}\dot{C}_{-\kappa_r}(C_\alpha + \hat{C}_{\alpha,-\kappa_t}C_{-\kappa_t})\Big). \tag{131}$$

Using (129) again, we subtract $(\kappa_s^2 - \beta^2)C_{-\kappa_s,\beta} = \dot{C}_{-\kappa_s}C_\beta - \dot{C}_\beta C_{-\kappa_s}$ to get

$$(\alpha^2 - \kappa_s^2)\hat{C}_{\alpha,-\kappa_s}C_{-\kappa_s,\beta} = \Big(\dot{C}_\alpha\hat{C}_{\beta,-\kappa_t}C_{-\kappa_t} + \hat{C}_{\alpha,-\kappa_r}\dot{C}_{-\kappa_r}(\quad \hat{C}_{\beta,-\kappa_t}C_{-\kappa_t})\Big)$$
$$- \Big(\quad\quad\quad\quad \hat{C}_{\beta,-\kappa_r}\dot{C}_{-\kappa_r}(C_\alpha + \hat{C}_{\alpha,-\kappa_t}C_{-\kappa_t})\Big). \tag{132}$$

Similarly, we can rewrite $\hat{C}_{\alpha,-\kappa_s} = C_{\alpha,-\kappa_t}(\mathcal{C})_{-\kappa_t,-\kappa_s}$ and subtract the identity (129) for $(\alpha^2 - \kappa_t^2)C_{\alpha,-\kappa_t}$, resulting in

$$(\kappa_t^2 - \kappa_s^2)C_{\alpha,-\kappa_t}(\mathcal{C})_{-\kappa_t,-\kappa_s}C_{-\kappa_s,\beta} = \hat{C}_{\alpha,-\kappa_t}\hat{C}_{\beta,-\kappa_r}\Big[\dot{C}_{-\kappa_t}C_{-\kappa_r} - \dot{C}_{-\kappa_r}C_{-\kappa_t}\Big]. \tag{133}$$

Finally, using (129) once more with $\dot{C}_{-\kappa_t}C_{-\kappa_r} - \dot{C}_{-\kappa_r}C_{-\kappa_t} = (\kappa_r^2 - \kappa_t^2)C_{\kappa_r,\kappa_t}$ leads to

$$(\kappa_t^2 - \kappa_s^2)C_{\alpha,-\kappa_t}(\mathcal{C})_{-\kappa_t,-\kappa_s}C_{-\kappa_s,\beta} = (\kappa_t^2 - \kappa_s^2)\hat{C}_{\alpha,-\kappa_t}C_{\kappa_t,\kappa_s}\hat{C}_{\beta,-\kappa_s} \,, \tag{134}$$

which is true and can be seen directly by plugging in the definitions of $\hat{C}_{\alpha,-\kappa_\ell}$ and $\mathcal{C}_{-\kappa_t,-\kappa_s}$ and using (126).

## A.3 The second equation

Here we consider the differential equation (34). The trans-series from (57) can be written in terms of $\hat{C}$ as

$$\chi_\alpha = \frac{1}{2}G_+(i\alpha)\hat{C}_\alpha + \frac{1}{2}G_-(i\alpha)\hat{C}_{-\alpha} \,. \tag{135}$$

Further, we rewrite the function $F$ as

$$F = f + \hat{f} \,, \tag{136}$$

where $f$ is the perturbative part and $\hat{f}$ is the non-perturbative part.

We now turn to the third differential equation (34), which can be written in terms of the $\hat{C}$'s as

$$(\ddot{\hat{C}}_\alpha + \sigma_\alpha^+ \ddot{\hat{C}}_{-\alpha}) - \alpha^2(\hat{C}_\alpha + \sigma_\alpha^+ \hat{C}_{-\alpha}) = (f + \hat{f})(\hat{C}_\alpha + \sigma_\alpha^+ \hat{C}_{-\alpha}) \,. \tag{137}$$

---

[14]For sake of readability, we leave the sums here implicit and also drop factors of $S_{-\kappa_\ell}$. Whenever there are two $C$'s with the same index $-\kappa_\ell$ a summation $\sum_{\ell=1}^\infty S_{-\kappa_\ell}$ including the factor $S_{-\kappa_\ell}$ should be inserted.

The expression above once again must hold for each combination of prefactors $\sigma_\alpha^+$ and hence it will be sufficient to consider the equation

$$\ddot{C}_\alpha + \frac{\mathrm{d}^2}{\mathrm{d}b^2}(C_{\alpha,-\kappa_r}(\mathcal{C})_{-\kappa_r,-\kappa_s}C_{-\kappa_s}) - \alpha^2(C_\alpha + C_{\alpha,-\kappa_r}(\mathcal{C})_{-\kappa_r,-\kappa_s}C_{-\kappa_s})$$
$$= (f + \hat{f})[C_\alpha + C_{\alpha,-\kappa_r}(\mathcal{C})_{-\kappa_r,-\kappa_s}C_{-\kappa_s}], \quad (138)$$

in the following. We can eliminate $f$ from the right hand side by using (121), leaving us with

$$\left[\frac{\mathrm{d}^2}{\mathrm{d}b^2}(C_{\alpha,-\kappa_r}(\mathcal{C})_{-\kappa_r,-\kappa_s})\right]C_{-\kappa_s} + 2\left[\frac{\mathrm{d}}{\mathrm{d}b}(C_{\alpha,-\kappa_r}(\mathcal{C})_{-\kappa_r,-\kappa_s})\right]\dot{C}_{-\kappa_s}$$
$$+ (\kappa_s^2 - \alpha^2)C_{\alpha,-\kappa_r}(\mathcal{C})_{-\kappa_r,-\kappa_s}C_{-\kappa_s} = \hat{f}[C_\alpha + C_{\alpha,-\kappa_r}(\mathcal{C})_{-\kappa_r,-\kappa_s}C_{-\kappa_s}]. \quad (139)$$

Performing the differentiation and using the identity from (127) leads to

$$\left[\dot{\hat{C}}_\alpha\hat{C}_{-\kappa_s} + \hat{C}_\alpha\dot{\hat{C}}_{-\kappa_s}\right]S_{\kappa_s}C_{-\kappa_s} + 2\hat{C}_\alpha\hat{C}_{-\kappa_s}S_{\kappa_s}\dot{C}_{-\kappa_s} + (\kappa_s^2 - \alpha^2)\hat{C}_{\alpha,-\kappa_s}S_{\kappa_s}C_{-\kappa_s}$$
$$= \hat{f}[C_\alpha + C_{\alpha,-\kappa_s}(\mathcal{C})_{-\kappa_s,-\kappa_s}C_{-\kappa_s}]. \quad (140)$$

Finally, we use (128) to rewrite $(\kappa_s^2 - \alpha^2)\hat{C}_{\alpha,-\kappa_s} = \hat{C}_{-\kappa_s}\dot{\hat{C}}_\alpha - \hat{C}_\alpha\dot{\hat{C}}_{-\kappa_s}$. The final result reads

$$\left[2S_{\kappa_s}\left(\dot{\hat{C}}_{-\kappa_s}C_{\kappa_s} + \hat{C}_{-\kappa_s}\dot{C}_{\kappa_s}\right)\right]\hat{C}_\alpha = \hat{f}\hat{C}_\alpha, \quad (141)$$

from which we can read off the solutions for $\hat{f}$ as

$$\hat{f} = \sum_{s=1}2S_{\kappa_s}\left(\dot{\hat{C}}_{-\kappa_r}C_{-\kappa_s} + \hat{C}_{-\kappa_s}\dot{C}_{-\kappa_s}\right)$$
$$= \sum_{s=1}2S_{\kappa_s}\frac{\mathrm{d}}{\mathrm{d}B}\hat{C}_{-\kappa_s}C_{-\kappa_s}. \quad (142)$$

Hence, the non-perturbative part $\hat{f}$ of $F$ can be expressed in terms of the same building blocks as the observables.

## A.4   The moments

Recall the differential equation relating moments $\phi_\ell$ to $\phi_{\ell-1}$ from (35), which we can also write as

$$\ddot{\phi}_\ell - 2\frac{\dot{\hat{C}}_0}{\hat{C}_0}\dot{\phi}_\ell = 2\ell(2\ell - 1)\pi^{-2}\phi_{\ell-1}. \quad (143)$$

The proposed trans-series for the moments (63) takes then the form

$$\phi_\ell = \phi_\ell^{(0)} + \frac{4}{\pi^{2\ell+1}}\sum_{s=1}^\infty \hat{C}_{0,-\kappa_s}C_{-\kappa_s;\ell}, \quad (144)$$

where we introduced $C_{\alpha;\ell} = e^{\alpha B}A_{\alpha;\ell}$. At the perturbative order, (143) takes the form

$$\ddot{\phi}_\ell^{(0)} - 2\frac{\dot{C}_0}{C_0}\dot{\phi}_\ell^{(0)} = 2\ell(2\ell - 1)\pi^{-2}\phi_{\ell-1}^{(0)}. \quad (145)$$

A similar equation can be obtained for the generalised moments from (64) and reads

$$\ddot{\phi}_{\alpha;\ell} - 2\frac{\dot{\chi}_\alpha}{\chi_\alpha}\dot{\phi}_{\alpha;\ell} + \alpha^2\phi_{\alpha;\ell} = 2\ell(2\ell - 1)\pi^{-2}\phi_{\alpha;\ell-1}. \quad (146)$$

Considering only the perturbative part of the equation above we can find the relation for the $C_{\alpha,\ell}$ given as

$$\ddot{C}_{\alpha;\ell} - 2\frac{\dot{C}_\alpha}{C_\alpha}\dot{C}_{\alpha;\ell} + \alpha^2 C_{\alpha;\ell} = 2\ell(2\ell-1)C_{\alpha;\ell-1}\,. \tag{147}$$

Finally, we have (36) relating the generalised moments. Restricting to its perturbative part we have

$$\dot{C}_{\alpha;\ell} = \frac{C_\alpha}{C_0}\dot{A}_{0;\ell}\,. \tag{148}$$

We will now show that the trans-series ansatz for the moments solves the corresponding differential equation by substituting (144) into (143). From this, we obtain

$$
\begin{aligned}
(C_0 + \hat{C}_{0,-\kappa_s}C_{-\kappa_s})\partial_B^2(A_{0;\ell} + \hat{C}_{0,-\kappa_r}C_{-\kappa_r;\ell}) & \\
- 2[\partial_B(C_0 + \hat{C}_{0,-\kappa_s}C_{-\kappa_s})][\partial_B(A_{0;\ell} + \hat{C}_{0,-\kappa_r}C_{-\kappa_r;\ell})] & \\
= 2\ell(2\ell-1)(C_0 + \hat{C}_{0,-\kappa_s}C_{-\kappa_s})(A_{0;\ell-1} + \hat{C}_{0,-\kappa_r}C_{-\kappa_r;\ell-1}) &
\end{aligned} \tag{149}
$$

We can remove the purely perturbative part of the expression above using (145). This yields

$$
\begin{aligned}
\hat{C}_{0,-\kappa_s}C_{-\kappa_s}\ddot{A}_{0;\ell} + \hat{C}_0\partial_B^2\hat{C}_{0,-\kappa_s}C_{-\kappa_s;\ell} - 2\dot{\hat{C}}_0\partial_B\hat{C}_{0,-\kappa_s}C_{-\kappa_s;\ell} - 2(\partial_B\hat{C}_{0,-\kappa_s}C_{-\kappa_s})\dot{A}_{0;\ell} & \\
= 2\ell(2\ell-1)\hat{C}_0\hat{C}_{0,-\kappa_s}C_{-\kappa_s;\ell-1} + 2\ell(2\ell-1)\hat{C}_{0,-\kappa_s}C_{-\kappa_s}A_{0;\ell-1}\,. &
\end{aligned} \tag{150}
$$

Next, we use (145) again with $\hat{C}_{0,-\kappa_s}C_{-\kappa_s}\ddot{A}_{0;\ell} = \hat{C}_{0,-\kappa_s}C_{-\kappa_s}[2\ell(2\ell-1)A_{0;\ell-1} + 2\frac{\dot{C}_0}{C_0}\dot{A}_{0;\ell}]$. This allows us to write

$$
\begin{aligned}
2\hat{C}_{0,-\kappa_s}C_{-\kappa_s}\frac{\dot{C}_0}{C_0}\dot{A}_{0;\ell} + \hat{C}_0\partial_B^2\hat{C}_{0,-\kappa_s}C_{-\kappa_s;\ell} - 2\dot{\hat{C}}_0\partial_B\hat{C}_{0,-\kappa_s}C_{-\kappa_s;\ell} - 2(\partial_B\hat{C}_{0,-\kappa_s}C_{-\kappa_s})\dot{A}_{0;\ell} & \\
= 2\ell(2\ell-1)\hat{C}_0\hat{C}_{0,-\kappa_s}C_{-\kappa_s;\ell-1}\,. &
\end{aligned} \tag{151}
$$

We can now execute the differentiations with respect to $B$ using the identities from Section A.1. The result reads

$$
\begin{aligned}
\hat{C}_0\Big(\dot{\hat{C}}_0\hat{C}_{-\kappa_s}C_{-\kappa_s;\ell} + \hat{C}_0\dot{\hat{C}}_{-\kappa_s}C_{-\kappa_s;\ell} + 2\hat{C}_0\hat{C}_{-\kappa_s}\dot{C}_{-\kappa_s;\ell} + \hat{C}_{0,-\kappa_s}\ddot{C}_{-\kappa_s;\ell}\Big) & \\
- 2\dot{\hat{C}}_0\Big(\hat{C}_0\hat{C}_{-\kappa_s}C_{-\kappa_s;\ell} + \hat{C}_{0,-\kappa_s}\dot{C}_{-\kappa_s;\ell}\Big) - 2(\partial_B\hat{C}_{0,-\kappa_s}C_{-\kappa_s})\dot{A}_{0;\ell} + 2\hat{C}_{0,-\kappa_s}C_{-\kappa_s}\frac{\dot{C}_0}{C_0}\dot{A}_{0;\ell} & \\
= 2\ell(2\ell-1)\hat{C}_0\hat{C}_{0,-\kappa_s}C_{-\kappa_s;\ell-1}\,. &
\end{aligned} \tag{152}
$$

Applying (147) for $\ddot{C}_{-\kappa_s;\ell}$, we obtain

$$
\begin{aligned}
0 = (\hat{C}_0)^2\Big[\dot{\hat{C}}_{-\kappa_s}C_{-\kappa_s;\ell} + 2\hat{C}_{-\kappa_s}\dot{C}_{-\kappa_s;\ell}\Big] + \hat{C}_0\hat{C}_{0,-\kappa_s}\left[2\frac{\dot{C}_{-\kappa_s}}{C_{-\kappa_s}}\dot{C}_{-\kappa_s;\ell} - \kappa_s^2 C_{-\kappa_s;\ell}\right] & \\
- \dot{\hat{C}}_0\hat{C}_0\hat{C}_{-\kappa_s}C_{-\kappa_s;\ell} - 2\dot{\hat{C}}_0\hat{C}_{0,-\kappa_s}\dot{C}_{-\kappa_s;\ell} - 2(\partial_B\hat{C}_{0,-\kappa_s}C_{-\kappa_s})\dot{A}_{0;\ell} + 2\hat{C}_{0,-\kappa_s}C_{-\kappa_s}\frac{\dot{C}_0}{C_0}\dot{A}_{0;\ell}\,. &
\end{aligned} \tag{153}
$$

Let us now consider the coefficient of $C_{-\kappa_s;\ell}$, which is given by

$$C_{-\kappa_s;\ell}\hat{C}_0\Big[\hat{C}_0\dot{\hat{C}}_{-\kappa_s} - \dot{\hat{C}}_0\hat{C}_{-\kappa_s} - \kappa_s^2\hat{C}_{0,-\kappa_s}\Big] = C_{-\kappa_s;\ell}\hat{C}_0\Big[\kappa_s^2\hat{C}_{-\kappa_s,0} - \kappa_s^2\hat{C}_{0,-\kappa_s}\Big] = 0\,, \tag{154}$$

where we used (128) to see that $C_{-\kappa_s;\ell}$ drops from (153). Further, using (36) we can substitute $\dot{A}_{0;\ell} = \frac{C_0}{C_{-\kappa_s}}\dot{C}_{-\kappa_s;\ell}$. Hence, (153) simplifies to

$$
\begin{aligned}
0 = 2\dot{C}_{-\kappa_s;\ell}\Bigg[&(\hat{C}_0)^2\hat{C}_{-\kappa_s} + \hat{C}_0\hat{C}_{0,-\kappa_s}\frac{\dot{C}_{-\kappa_s}}{C_{-\kappa_s}} - \dot{\hat{C}}_0\hat{C}_{0,-\kappa_s}\\
&- (\hat{C}_0\hat{C}_{-\kappa_s}C_{-\kappa_s} + \hat{C}_{0,-\kappa_s}\dot{C}_{-\kappa_s})\frac{C_0}{C_{-\kappa_s}} + \hat{C}_{0,-\kappa_s}C_{-\kappa_s}\frac{\dot{C}_0}{C_0}\frac{C_0}{C_{-\kappa_s}}\Bigg]\\
= 2C_{-\kappa_s;\ell}\Bigg[&\left(\hat{C}_0 - C_0\right)\left(\hat{C}_0\hat{C}_{-\kappa_s} + \hat{C}_{0,-\kappa_s}\frac{\dot{C}_{-\kappa_s}}{C_{-\kappa_s}}\right) - \left(\dot{\hat{C}}_0 - \dot{C}_0\right)\hat{C}_{0,-\kappa_s}\Bigg]\\
= 2C_{-\kappa_s;\ell}\Bigg[&\left(\hat{C}_{0,-\kappa_r}C_{-\kappa_r}\right)\left(\hat{C}_0\hat{C}_{-\kappa_s} + \hat{C}_{0,-\kappa_s}\frac{\dot{C}_{-\kappa_s}}{C_{-\kappa_s}}\right)\\
&- \left(\hat{C}_0\hat{C}_{-\kappa_r}C_{-\kappa_r} + \hat{C}_{0,-\kappa_r}\dot{C}_{-\kappa_r}\right)\hat{C}_{0,-\kappa_s}\Bigg],
\end{aligned}
\tag{155}
$$

where we used $\hat{C}_0 - C_0 = \hat{C}_{0,-\kappa_r}C_{-\kappa_r}$ and $\dot{\hat{C}}_{0,-\kappa_r} = \hat{C}_0\hat{C}_{-\kappa_r}$. Reordering the terms and exchanging $r \leftrightarrow s$ in the latter part of the expression results in

$$
\begin{aligned}
0 = 2\Bigg[&\hat{C}_0(C_{-\kappa_s;\ell}C_{-\kappa_r} - C_{-\kappa_r;\ell}C_{-\kappa_s})\hat{C}_{0,-\kappa_r}\hat{C}_{-\kappa_s}\\
&+ \hat{C}_{0,-\kappa_r}\hat{C}_{0,-\kappa_s}\left(C_{-\kappa_s;\ell}\frac{C_{-\kappa_r}}{C_{-\kappa_s}} - C_{-\kappa_r;\ell}\right)\dot{\hat{C}}_{-\kappa_s}\Bigg].
\end{aligned}
\tag{156}
$$

Using a generalisation of (148) we can see, that the terms in brackets indeed vanish, since

$$
C_{-\kappa_s;\ell}\frac{C_{-\kappa_r}}{C_{-\kappa_s}} = C_{-\kappa_r;\ell}\,.
\tag{157}
$$

Therefore we conclude that the trans-series ansatz (144) is indeed a solution to the differential equation for the moments (143).

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
