# Peer review of "The complete trans-series for conserved charges in the Lieb-Liniger model"

_SciPost Physics_

## Round 2 · Referee Report · Anonymous (Referee 1) · 2025-7-15

Strengths

1- Opens new direction in low-energy integrable models; 2- Provides a recipe how to construct the nonperturbative series expansion of the moments of the rapidity distribution in the Lieb-Liniger model.

Weaknesses

1- For some readers a weakness is that the solution is rather involved; however in my opinion this should not be understood in negative context but as an inconvenience.

Report

The authors study the structure and connections between the perturbative and nonperturbative parts of the series for the moments of the rapidity distribution in the Lieb-Liniger model at weak interactions. Using and extending some exact relations between the moments derived by Ristivojevic in Refs. [7] and [23], the authors extended the study to the nonperturbative part of the asymptotic series, which they eventually constructed. The agreement with very precise numerical algorithm to solve the original integral equation of Ref. [31] gives conclusive evidence that the goal of the paper is achieved, see Fig. 2 and Table 2. The manuscript is clearly written with great amount of details that interesting readers can follow. My opinion is that the present paper will become a cornerstone for future studies of nonperturbative effects in the Lieb-Liniger model.

Requested changes

1- Could the authors provide a physical understanding of exponentially suppressed terms in the ground-state energy, for example? It would be great if a parallel/comparison can be drawn with the findings of the paper of Marino https://doi.org/10.1088/1742-5468/ab4802 where the interpretation of the nonperturbative part is physical.

2- Typography: there is small inconsistency with dashes and hyphens: Lieb--Liniger vs Lieb-Liniger, Bose-Einstein, Coleman-Mermin-Wagner, etc. transseries vs trans-series.

Recommendation

Ask for minor revision

---

## Editorial Decision

resubmitted